

# Stopping the Flood: Could We Use Targeted Geoengineering to Mitigate Sea Level Rise?

Michael J. Wolovick[1] and John C. Moore[2,3]

[1]Atmosphere and Ocean Sciences Program, Department of Geosciences, Princeton University, GFDL, 201 Forrestal Road, Princeton, NJ 08540, USA
[2]College of Global Change and Earth System Science, Beijing Normal University, Beijing, China
[3]Arctic Centre, University of Lapland, Finland

**Correspondence:** M.J. Wolovick (wolovick@princeton.edu)

**Abstract.** The Marine Ice Sheet Instability (MISI) is a dynamic feedback that can cause an ice sheet to enter a runaway collapse. Thwaites Glacier, West Antarctica, is the largest individual source of future sea level rise and may have already entered the MISI. Here, we use a suite of coupled ice–ocean flowband simulations to explore whether targeted geoengineering using an artificial sill or artificial ice rises could counter a collapse. Successful interventions occur when the floating ice shelf regrounds on the pinning points, increasing buttressing and reducing ice flux across the grounding line. Regrounding is more likely with a
continuous sill that is able to block warm water transport to the grounding line. The smallest design we consider is comparable in scale to existing civil engineering projects but has only a 30% success rate, while larger designs are more effective. There are multiple possible routes forward to improve upon the designs that we considered, and with decades or more to research designs it is plausible that the scientific community could come up with a plan that was both effective and achievable. While reducing emissions remains the short-term priority for minimizing the effects of climate change, in the long run humanity may
need to develop contingency plans to deal with an ice sheet collapse.

## 1 Introduction

Human emissions of carbon dioxide are altering the Earth's climate in ways that are likely to have long-lasting consequences
for both human societies and natural ecosystems (IPCC, 2013). Emissions cuts promised by existing national commitments are insufficient to achieve the 2°C goal set by the international Paris Agreement (UNEP, 2016). Geoengineering, in the form of either carbon removal or solar radiation management, has been proposed as a method to close this gap (Shepherd et al., 2009). Carbon removal, or "negative emissions", is a set of methods to remove $CO_2$ from the atmosphere and sequester it either in the ground or in the deep ocean (Shepherd et al., 2009). Solar radiation management is a method to limit the rise in global
temperature by increasing the planetary albedo and reflecting more sunlight back to space, for example by injecting aerosols





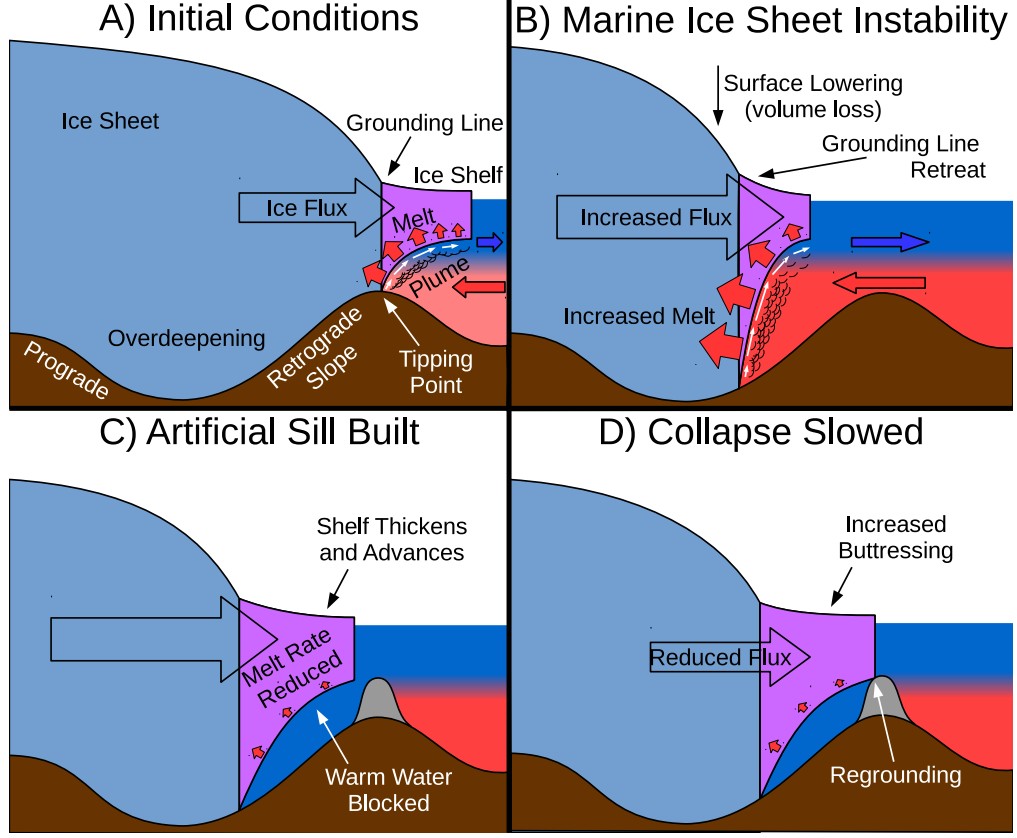

**Figure 1.** Schematic diagram of marine ice sheet instability and mitigation with an artificial sill. Brown represents bedrock, light blue represents grounded ice sheet, purple represents floating ice shelf, and gray represents an artificial sill. Ocean temperatures are drawn to represent the typical stratification faced by marine-terminating ice streams: warm salty water at depth and cold fresh water near the surface.

into the stratosphere (Shepherd et al., 2009). Solar radiation management has been extensively studied in the GeoMIP6 project (Kravitz et al., 2015), but its effect on the ice sheets remains unknown (Irvine et al., 2018).

Instead of trying to modify the entire climate, humanity could employ a locally targeted intervention aimed at specific high-leverage locations such as ice streams and outlet glaciers (Moore et al., 2018). Here, we explore whether it could be possible to

5   use either a continuous artificial sill or isolated artificial ice rises to counteract the Marine Ice Sheet Instability (MISI)(Fig 1). The MISI is a dynamic feedback that can cause an ice sheet to rapidly collapse due to a runaway retreat of the grounding line, the point at which the ice lifts off the bedrock and goes afloat on the ocean. Ice sheets are vulnerable to the MISI when their grounding line is located on a retrograde bed, meaning that the base slopes down towards the center of the ice sheet (Hughes, 1973; Weertman, 1974; Thomas and Bentley, 1978; Mercer, 1978; Schoof, 2007).

10   The instability operates as follows: as the grounding line retreats down a retrograde bed, the ice thickness at the grounding line increases, and ice flux across the grounding line increases strongly with local ice thickness (Schoof, 2007). As flux across





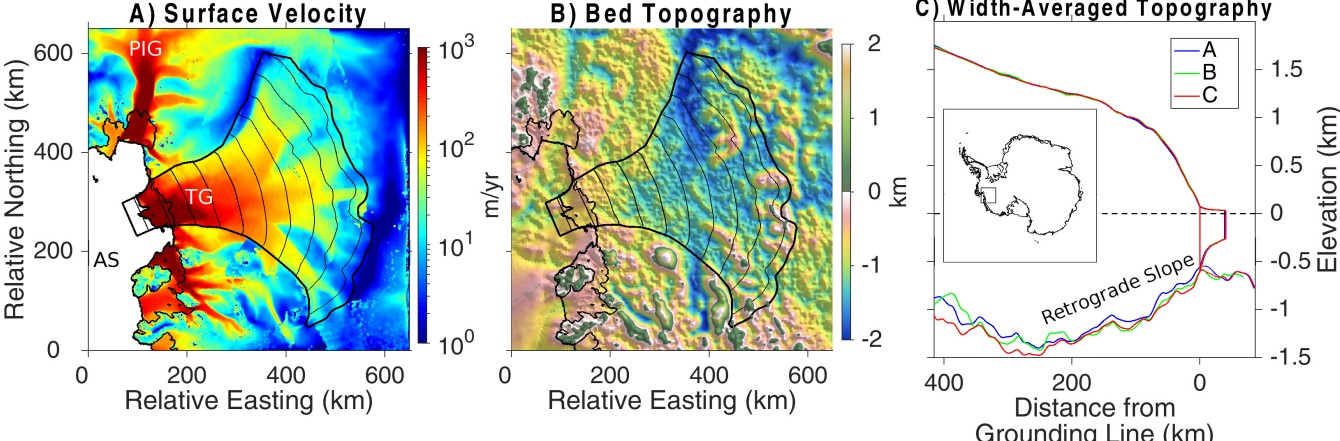

**Figure 2.** The Amundsen Sea sector of West Antarctica. A) Surface velocity (observations (Rignot et al., 2011) merged with balance velocity), B) bed topography (Fretwell et al., 2013), and C) width-averaged bed and surface profiles for Thwaites Glacier. AS=Amundsen Sea, PIG=Pine Island Glacier, TG=Thwaites Glacier. Inset shows location within Antarctica. Overlay lines show wide flowband boundaries and 50 km contours of along–flow distance, as well as grounding line and calving front. Three seperate methods were used to compute the width-averaged topography in (C) (see S1.3). Note severely overdeepened bed geometry in (B) and (C). Vertical exaggeration in (C) is 150.

the grounding line increases so does the rate of stretching and thinning, leading to further grounding line retreat (Hughes, 1973; Weertman, 1974; Thomas and Bentley, 1978; Mercer, 1978; Schoof, 2007). In the canonical 1D treatment of the problem, a grounding line on a retrograde slope is unconditionally unstable (Schoof, 2007). Stable grounding lines on retrograde slopes require complicating factors such as lateral buttressing, variable basal drag, or gravitational effects (Gudmundsson et al., 2012; Robel et al., 2016; Gomez et al., 2010). The initiation of the MISI is especially sensitive to basal melting caused by the presence of warm ocean waters near the grounding line (Joughin et al., 2012). Some authors have suggested that encroaching warm water has already triggered the MISI in the Amundsen Sea sector of West Antarctica(Joughin et al., 2014; Favier et al., 2014; Rignot et al., 2014), including at Pine Island and Thwaites Glaciers (Fig 2).

The hypothesis that the MISI has already been triggered in the Amundsen sector is consistent both with the available data and with glaciological theory, but the available data (mostly) begin in the 1990's (e.g. Shepherd et al., 2012). We regard this hypothesis to be probable but not yet proven, and we proceed with the understanding that the probability of an ice sheet collapse need not be 100% for the risk to be an important societal concern. There is also uncertainty about whether the ocean forcing that (may have) pushed the ice sheet over the edge was caused by human activity (Steig et al., 2012). We proceed with the understanding that the societal consequences of a collapse will be the same regardless of whether or not humanity is responsible.

Without extensive investments in dikes, levees, and other coastal protection infrastructure, a sea level rise of 0.6–1.2 m in 2100 would produce US$50 trillion/yr in economic losses, temporary population displacements of 100-500 million people per year due to flooding, permanent depopulation of many coastal communities, and widespread loss of wetland ecosystems





(Hinkel et al., 2014; Jevrejeva et al., 2016). The coastal protection infrastructure required to prevent (most of) that destruction would itself cost US\$27-71 billion every year to build, maintain, and upgrade (Hinkel et al., 2014).

And yet those figures are based on much less sea level rise than the 3.4 or 19 m that would result from a collapse of the marine–based portions of West or East Antarctica, respectively (Fretwell et al., 2013). Glaciologists believe that this much sea
level rise will probably not occur by 2100 (Bamber and Aspinall, 2013), but only because most models predict that it will take until the 22[nd] or 23[rd] centuries for a collapse to reach full speed (DeConto and Pollard, 2016; Winkelmann et al., 2015; Golledge et al., 2015). Once a collapse reaches full speed, sea level rise rates of several meters per century are common in modern models (DeConto and Pollard, 2016; Winkelmann et al., 2015; Golledge et al., 2015), consistent with geological evidence that sea level rose at 4.1–5.3 meters per century during Earth's last deglaciation (Deschamps et al., 2012). It is unknown if traditional coastal
protection could keep up with such a rapid rate of worldwide sea level rise, and such rapid sea level rise would probably be just as harmful to society in 2200 or 2300 as it would be in 2100, so targeted geoengineering could be a cost-effective adaptation strategy.

## 2   Proposal

Here, we explore the possibility of using either a continuous artificial sill or isolated artificial ice rises to counter the MISI
(Moore et al., 2018). We explore the effect of this intervention on the largest ice stream for which the MISI may have already been triggered (Joughin et al., 2014): Thwaites Glacier, West Antarctica, since if it works there then we would expect it to work on less challenging glaciers as well. Standard theory suggests that, once initiated, the MISI must continue until the grounding line retreats onto a prograde slope or the ice sheet completely collapses (Schoof, 2007). The question that we seek to answer is: can an ongoing collapse be slowed or reversed by modifying the bathymetry in front of the glacier? And how much must
the bathymetry be modified to achieve that goal?

We envision both the sill and the ice rises as extremely simple structures, merely piles of aggregate on the ocean floor, although more advanced structures could certainly be explored in the future. We use a reduced complexity ice/ocean model to investigate the effectiveness of several different designs, and to explore how the effectiveness is reduced when some or all of the warm water is allowed to bypass the sill. We then discuss how these model results translate into rough design requirements
for a successful intervention.

## 3   Methods

We use the least complex model that can address this question: for the ice, we use a flowband model with parameterized lateral buttressing (S1.1), while for the ocean we use a model of the turbulent buoyant plume at the ice base (S1.2) following Jenkins' model (Jenkins, 1991, 2011). We used multiple width–averaging schemes to produce our flowband profiles (S1.3) and then
inverted the surface velocity data long those profiles to get basal drag (S1.4). The inversion was performed for a linear sliding rule, and the inverted drag coefficient was split into spatially variable velocity and stress scales, $u_0(x)$ and $\tau_0(x)$, in order to





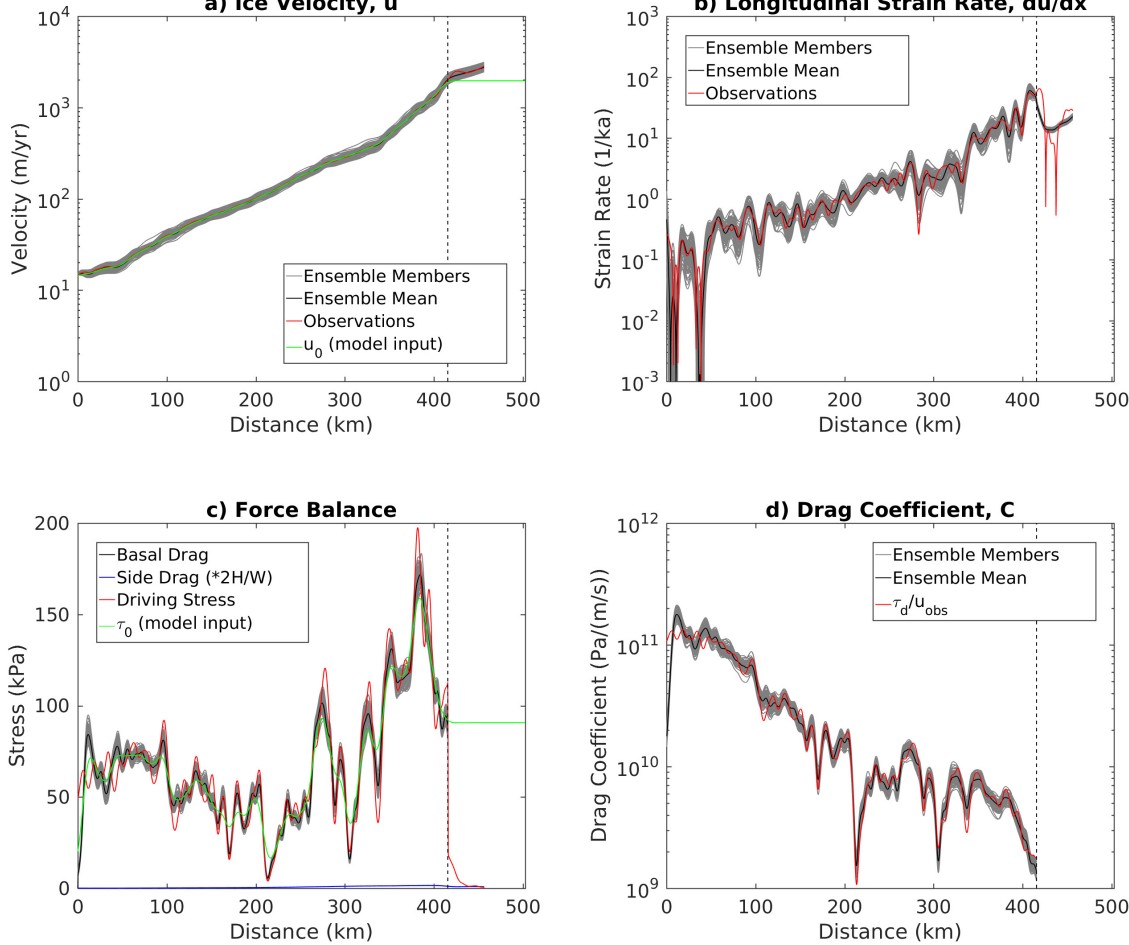

**Figure 3.** Velocity inversion for the C flowband (wide boundaries with flux–weighted averaging). a) Velocity, b) strain rate, c) stress, and d) drag coefficient. Gray lines show individual members of the final population of models within the inversion (S1.4), black lines show the ensemble mean, red lines show observations, and green lines show the quantities used as inputs to the model sliding rule. Vertical dashed line represents the grounding line; note that $\tau_0$ and $u_0$ were extrapolated beyond the grounding line to allow the glacier to advance.

allow subsequent experiments to change the slip exponent without rerunning the inversion. An example of the inversion results is shown in Figure 3. We also ran resolution tests of our model to ensure that it could accurately capture both the steady–state and the transient dynamics of the MISI (S2).





## 3.1  Experiment Description

We assembled a sample of model experiments in sets of three. We defined the flowband in three different ways, described in supplementary section 1.3, to sample the uncertainty associated with using a reduced-dimension flowband model. We explored three separate calving laws and three separate sliding exponents, in order to sample the uncertainty associated with different parameterizations of physical processes. We used values of the sliding exponent of 1, 3, and 10, in order to capture a range of ice-bed properties from viscous to plastic. All of the sliding laws had the same form, namely, a power-law relationship between shear stress and slip rate (Equation S6). For iceberg calving, on the other hand, we used different functional forms reflecting different variables that could plausibly impact calving: thickness, velocity, and melt rate. Finally, we used three forcing scenarios: constant climate control runs, climate warming runs, and climate warming runs with geoengineering. The sill was rapidly built beginning 100 years into the 1000 year model runs, with construction lasting 10 years. We ran experiments with four different designs: a tall sill built in the open bay, and a short sill built on the present–day grounding line that blocked either 100%, 50%, or 0% of the warm ocean water. We took the experiment with 0% water blockage to represent isolated ice rises instead of a continuous sill. We limited the last three scenarios to the wide flowbands because the narrow flowbands did not enter a runaway collapse (see Results) and we were interested in the question of whether an ongoing collapse could be stopped. For the 50% blockage experiment, the ocean properties forcing the sill model were a linear combination of the properties at the sill top and the far–field stratification. The sill was not erodible and had the same sliding properties as were extrapolated to the rest of the ungrounded region (Fig 3). Overall, we performed 135 model runs, 81 of which of which tested some version of an intervention.

## 3.2  Calving

For iceberg calving, we use one of three calving laws,

$$\dot{c}(H) = u_0 \frac{H_0}{H}, \tag{1}$$

$$\dot{c}(u,H) = u \frac{H_0}{H}, \text{and} \tag{2}$$

$$\dot{c}(\dot{m},H) = u_0 \frac{H_0 \dot{m}}{\dot{m}_0 H}, \tag{3}$$

where $\dot{c}$ is the calving rate, $u$ is ice velocity, $H$ is ice thickness, and $\dot{m}$ is frontal melt rate. Values with subscript 0 indicate constants set at the beginning of the model run, values without subscripts indicate model variables. The reference constants are taken from the present-day geometry (Fretwell et al., 2013) or frontal velocity (Rignot et al., 2011) of the glacier. For the melt-dependent calving rule (Equation 3), $m_0$ is taken from the geometric mean of the calving front melt rate in the first year of a model run with the front position held fixed. The melt-dependent calving rule is inspired by the melt-multiplier calving effect (O'Leary and Christoffersen, 2013). The velocity-dependent calving rule (Equation 2) is inspired by the known weakening effect of high ice velocity and associated high strain rates (e.g. Benn et al., 2007; Alley et al., 2008). All calving rules used an inverse-thickness dependence to prevent the formation of ice shelves that pinch out to zero thickness at their front. Without




increased calving rates for small ice thicknesses, early versions of the model often produced ice shelves that advanced and thinned until the front pinched out to zero thickness at the waterline. Real ice shelves and tidewater glaciers almost always terminate in a frontal cliff rather than pinching out to zero thickness (e.g. Fretwell et al., 2013; Morlighem et al., 2014). The price of this feature is that our model cannot include the marine ice cliff instability, which could play an important role in

accelerating West Antarctic collapse (DeConto and Pollard, 2016). However, this feature also guaranteed that our model never produced unphysically large ice cliffs in the first place, so in practice this was not an issue.

### 3.3 Climate Scenarios

For the constant climate scenario, we used present-day surface mass balance taken from the mean of two datasets (Arthern et al., 2006; Van de Berg et al., 2005) accessed through the ALBMAP compilation (Le Brocq et al., 2010) and width–averaged

onto our flowbands (S1.3). The plume model was forced by a piecewise linear stratification chosen to be similar to observations (Jacobs et al., 2011). The piecewise linear stratification is shown in Figure 4c.

For the warming scenario, we generated a schematic set of climate forcings loosely representing business as usual. Our goal was not to make a precise projection based on a specific IPCC climate scenario, but rather to capture the general features of climate warming as it affects the ice sheet in order to produce a baseline against which we could measure the performance of

the intervention. This approach to the forcings is similar to that taken by, for example, the SeaRISE project (Bindschadler et al., 2013). We included surface ablation at low altitudes, a mild increase in accumulation, and shoaling of the thermocline but no warming of deep ocean waters. All changes proceeded along an exponential approach to a new steady state, with an e-folding time of 200 years. The ultimate increase in accumulation rate was 10%, similar to the values found by climate models for the Amundsen Sea sector of West Antarctica under medium to high–end emissions scenarios (Bracegirdle et al., 2008). Surface

ablation was parameterized by a rising elevation profile; at the beginning of the model run the elevation of zero ablation was assumed to be sea level, and this elevation then rose and asymptotically approached a maximum value with the same 200 year e-folding time as the other climate changes. For an assumed 6°C of eventual warming in the Antarctic, comparable with estimates from climate models running high–end emissions scenarios (Bracegirdle et al., 2008), and a 7°C/km lapse rate, the height of zero ablation ultimately climbed 857 m. Below the elevation of no ablation, the ablation rate increased with a 1

m/yr/km lapse rate (Fig 4b). Ablation was confined to a 4 month summer ablation season. Surface melt from ablation was assumed to drain to the bed and flow to the grounding line, where it served as a boundary condition for the plume model of sub-shelf melt. Surface melt had no effect on basal sliding and no direct effect on iceberg calving, although an indirect effect existed for the melt-dependent calving law (Equation 3) because changes in freshwater forcing at the grounding line produce changes in the submarine melt rate at the calving front. The thermocline began between 700 m and 300 m, roughly following

observations (Jacobs et al., 2011), and was assumed to finish between 400 m and 100 m. Climate model simulations of the Southern Ocean are known to be rather poor at present with large model spread over the coming century on even the sign of ocean forcing (Sun et al., 2016). Additionally, melt rates depend in practice on local grounded icebergs, ice shelves, and sea ice (Cougnon et al., 2017).





**Figure 4.** Climate forcing used for the warming runs. Panel A) shows the asymptotically rising elevation of ablation, panel B) shows the annually averaged ablation profile below that elevation, and panels C) and D) show the shoaling CDW. The ablation rate in any year is computed by taking the elevation of zero ablation in that year (black dot in (A) and (B)) and applying the lapse rate shown in (B). Ablation is only applied during a four–month summer ablation season, so the instantaneous ablation rate is higher than the annual average shown in (B). Net surface mass balance is the sum of the vertically variable ablation rate and the horizontally variable accumulation rate (not shown).

For the geoengineering scenarios, we used the same climate forcing as the warming scenarios, but added an artificial sill after 100 years. The sill height increased linearly over a 10 year construction interval. When the sill blocked 100% of the



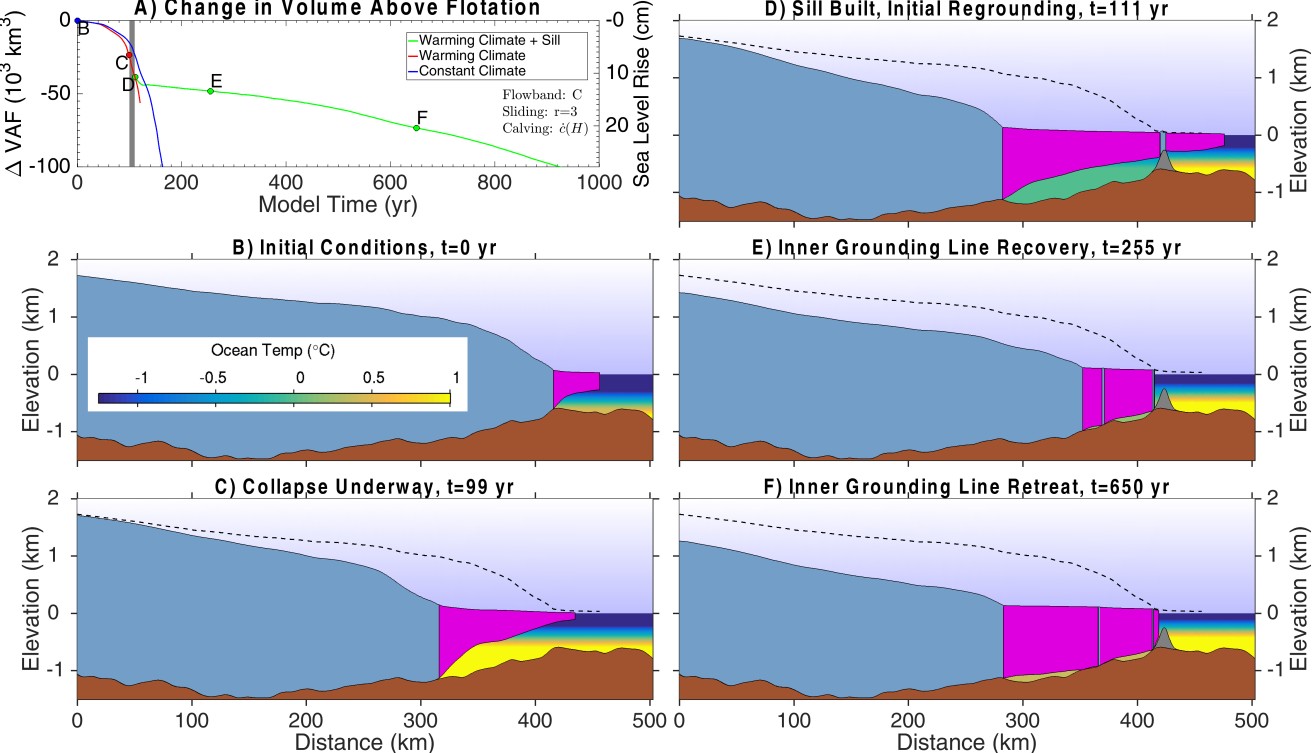

**Figure 5.** Example model output for an intervention that regrounded Thwaites glacier and slowed (but not reversed) sea level rise. The sill blocked 50% of the warm water in this scenario. A) Time series of volume above flotation and equivalent sea level rise. Vertical gray bars show when the sill was built. Snapshots (B-F) show model geometry and ocean temperature. Brown regions represent bedrock, pale blue represents grounded ice, purple represents floating ice shelf, and gray represents the sill. Dashed lines represent initial ice surface. Vertical exaggeration is 50.

warm water, then ocean properties at the sill top were assumed to overflow and fill the basin behind the sill. For lower blocking percentages, the water properties behind the sill were a linear combination of the far–field stratification and the water properties at the sill top. We used a Gaussian sill profile with a $2\sigma$ width of 7.5 km (tall sill) or 5 km (short sill) to ensure that the sill was smooth relative to the model grid size. The model results presented in this paper were run with a nominal grid size of 500 m
5    and a timestep of 0.02 yr.

## 4   Results

Under constant climate forcing, 50% of the experiments that we performed on Thwaites Glacier experienced a runaway marine ice sheet collapse within 1000 years. In the warming scenario, that number increased to 70% (Animation 1)– and all the exceptions were using a narrow flowband. Of the model experiments that represented Thwaites with wide flowband boundaries,




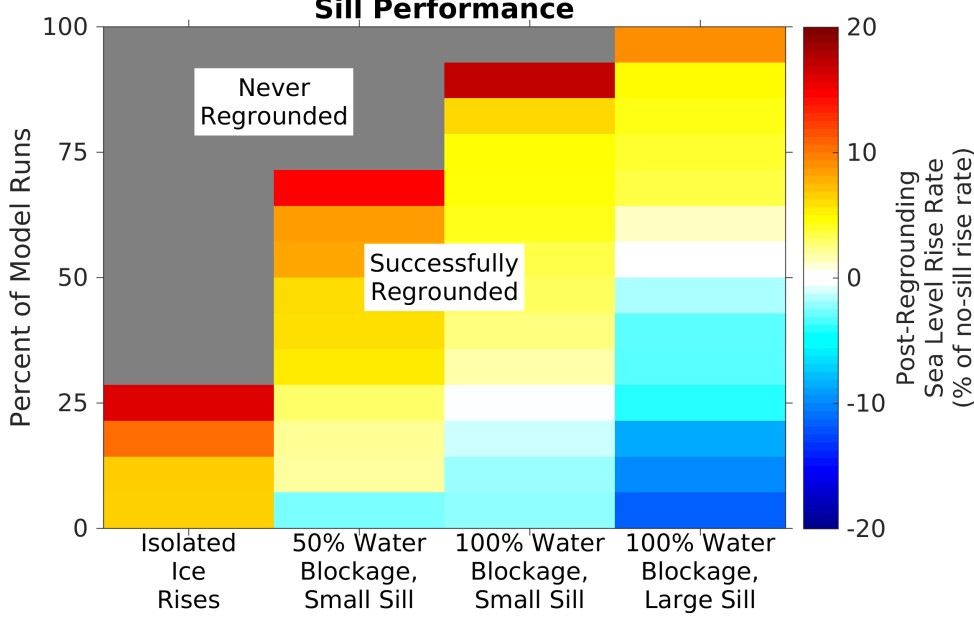

**Figure 6.** Summary of sill performance as a function of design scenario. "Isolated Ice Rises" are represented in the model by a small sill that blocks 0% of the warm water. Results are only shown for model runs that had entered a collapse before the sill was built. Gray areas represent model glaciers that never regrounded and continued a runaway collapse. Color areas represent model runs that successfully regrounded. Color value represents the rate of sea level rise after regrounding, expressed as a percentage of the sea level rise rate without a sill.

80% collapsed in a constant climate and 100% collapsed in a warming climate. Because we are interested in the question of whether an ongoing collapse can be mitigated, we limit our analysis to the wide flowbands in the rest of the paper. Our results are consistent with the hypothesis that the MISI has already been triggered for Thwaites, but they also suggest that the probability of collapse depends both on the climate forcing and on poorly known sliding and calving processes. The exact

5   timing and rate of the collapse varied between model experiments, but in general it was slow for the first century as the grounding line retreated the first 50-100 km from the present-day position and then accelerated as the grounding line moved onto weaker (Fig 3) and deeper (Fig 2) bed further inland. In some experiments the model grounding line had retreated as much as 150 km from its present–day position in the century before the sill was built.

   Regrounding of the floating ice shelf is key to the glacier's recovery from such a severely retreated position (Animations

10   2-5). As the grounding line retreats onto deeper bed, ice flux across the grounding line increases, removing mass from the grounded ice sheet and adding it to the floating shelf. If this mass input exceeds basal melt and frontal calving, then the shelf will thicken and flow outward. The thickened and lengthened shelf regrounds on the sill (Fig 5d). The initial regrounding splits the ice into a well-buttressed inland shelf and a seaward shelf with little buttressing. The seaward shelf is unprotected from melt or calving and thus shrinks over time, while the inland shelf thickens and regrounds (Fig 5e). In some experiments the





inland shelf completely regrounds and the glacier regains mass, while in others the innermost grounding line eventually starts retreating again and mass loss resumes, albeit at a lower rate than before (Fig 5a,f).

Both the odds of regrounding and the odds of mass gain were strong functions of the sill design (Fig 6). Isolated ice rises (represented in our model by a sill that blocked 0% of the warm water) successfully regrounded 30% of the time. A smaller sill worked 70% of the time if it could block half the warm water and 90% of the time if it could block all of it, and a larger sill that blocked all of the warm water worked 100% of the time. More effective designs were also more likely to regain mass after regrounding (Fig 6). As discussed below, the isolated ice rises are the only design that is a comparable scale to existing civil engineering projects, but our model results suggest a way that that design could be improved.

In some simulations, the ice shelf floated above the artificial pinning point but was too thin to touch down because of the high basal melt rate. If the ice shelf was locally thickened above the pinning point, it might reground and the intervention would be successful. Over 30 years ago, MacAyeal (MacAyeal, 1983) proposed that artificial ice rises could be created in the Ross Ice Shelf by pumping seawater onto the surface in the winter, so that it would freeze in place and thicken the shelf from above. More recently, Frieler and others (Frieler et al., 2016) proposed a similar scheme at a larger scale to offset sea level rise by adding mass to the slow–flowing areas of East Antarctica. While seawater pumping at a large enough scale to directly offset sea level rise is impractical, seawater pumping as a targeted method to thicken specific key locations of an ice shelf could be more feasible. If we include the model runs where a thin ice shelf floated over the pinning point without regrounding, then the success rate for this design would double to 60%.

## 5  Cost and Feasibility

Estimating the monetary cost of a project that will not begin for a century or two is difficult. An accurate estimate would require making assumptions about technology, economy, and Antarctic logistics a century hence. While it is tempting to assume that the remoteness and harshness of Antarctica precludes a large civil engineering project, consider that the annual budget for the US military is $583 billion (OMB, 2017), while the logistical budget for the US Antarctic Program is only $270 million (NSF, 2017), a difference of over three orders of magnitude. If rapidly rising sea level made Antarctica a global priority, then investment in the continent could easily increase by several orders of magnitude even without accounting for future economic growth. Consider also the rapid expansion of Antarctic infrastructure that occurred in the half century between the "heroic age" and the 1957/8 International Geophysical Year. In 1902 the entirety of humanity's Antarctic infrastructure was a wood hut by the shore of McMurdo Sound; 60 years later, McMurdo Station installed a nuclear reactor (AP, 1960).

The simple designs we envisage here allow direct comparison with existing engineering projects. A line of four isolated ice rises requires 0.1-1.5 km$^3$ of aggregate to build, depending on the strength of the aggregate (Table 1). That is comparable to the 0.1 km$^3$ that was used to create Palm Jumeirah in Dubai (US\$12 billion), the 0.3 km$^3$ that was used to create Hong Kong International Airport (\$20 billion), or the 1.6 km$^3$ that was moved for China's South to North Water Diversion Project (\$80 billion). Continuous sills require one to two orders of magnitude more material than this (Table 1), but reward their increased difficulty with increased odds of success (Fig 6).





**Table 1.** Aggregate Volume Requirements

| Structure Description | Sill Length (km) | Water Depth (m) | Sill Depth (m) | Strong Volume (km$^3$) | Weak Volume (km$^3$) |
|---|---|---|---|---|---|
| Palm Jumeirah | | | | 0.10 | |
| Panama Canal | | | | 0.20 | |
| Hong Kong International Airport | | | | 0.30 | |
| Suez Canal | | | | 1.0 | |
| South to North Water Diversion Project | | | | 1.6 | |
| Jakobshavn[1] | | 265 | 150 | 0.0032 | 0.044 |
| Jakobshavn[2] | 5 | 265 | 150 | 0.066 | 0.25 |
| Helheim[2] | 7 | 550 | 200 | 0.86 | 3.2 |
| Kangerdlugssuaq[2] | 8.5 | 450 | 100 | 1.0 | 3.9 |
| Petermann[3] | 19 | 350 | 210 | 0.37 | 1.4 |
| Petermann[2] | 20 | 430 | 100 | 2.2 | 8.1 |
| Pine Island[1] | | 685 | 420 | 0.039 | 0.54 |
| Pine Island[3] | 40 | 685 | 420 | 2.8 | 10 |
| Pine Island[4] | 50 | 685 | 300 | 7.4 | 28 |
| Pine Island[5] | 50 | 685 | 100 | 17 | 64 |
| Thwaites[1] | | 545 | 250 | 0.11 | 1.5 |
| Thwaites[3] | 80 | 545 | 250 | 7.0 | 26 |
| Thwaites[4] | 120 | 600 | 300 | 11 | 40 |
| Thwaites[5] | 120 | 600 | 100 | 30 | 110 |

**Table 1.** 1) isolated ice rises (two for Jakobshavn and PIG, four for Thwaites); 2) Sill built in fjord mouth; 3) sill under shelf; 4) low sill in open bay; 5) tall sill in open bay. Volume calculations assume that the sill is shaped like a triangular prism defined by a fixed angle of repose. Ice rises assume a conical shape with the same angle of repose. "Strong Volumes" use an angle of repose of 45°, "Weak Volumes" use 15°. Note that the "length" of the sill is the cross-flow dimension of the glacier or fjord. All volumes have been rounded to two significant figures.

The key to improving our designs is therefore to figure out how to get higher performance from less material. Small ice rises presently stabilize huge areas of ice shelf (Fürst et al., 2015), so buttressing alone does not require the construction of very large structures. As discussed above, we could coordinate the construction of artificial pinning points from below with seawater pumping to thicken the ice shelf from above (MacAyeal, 1983; Frieler et al., 2016). We could also try to optimize the tradeoff

5 between warm water blocking, buttressing, and structure volume by using fully-coupled three-dimensional ice-ocean models and assimilated ice and ocean observational data. With knowledge of the route of ocean currents in the sub–ice cavity, it may be possible to get the water–blocking performance of a continuous sill with less material. We could also coordinate construction with an atmospheric intervention designed to remove warm water from the sub–ice cavity by producing downwelling–favorable





winds in the Amundsen Sea, although adding an atmospheric intervention to a targeted geoengineering project may make it harder to keep the side effects confined to a local area. More fancifully, some have even proposed using large fiberglass curtains to block ocean currents (Cathcart et al., 2011), which we could use to block warm water transport between the pinning points.

Regardless of what design we ultimately choose, it would be prudent if humanity attempted smaller glaciers first in order to develop technology, prove the concept, and gain experience with attempting to manage ice dynamics. For example, at Jakobshavn Glacier in Greenland, two isolated pinning points designed to jam the iceberg melange would only require 0.003-0.04 km$^3$ of material, while a sill that completely blocked the fjord mouth at 150 m depth would require 0.07-0.25 km$^3$ (Table 1), and most glaciers in Greenland are smaller than Jakobshavn. Humanity could approach the challenge of Thwaites by sequentially climbing the difficulty ladder of smaller glaciers. Each rung of the ladder will require several decades to master the new challenges that will undoubtedly appear as the scale increases.

## 6   Discussion

We are not advocating that glacial geoengineering be attempted any time soon. An ice sheet intervention today would be at the edge of human capabilities. It would be comparable to the largest civil engineering projects that humanity has ever attempted, it would be located in a much harsher environment than the ones in which those projects were built, and it would have only a 30% probability of success. What we are advocating instead is the beginning of an incremental process of design improvement. In Section 5, we suggested multiple possible routes forward to improve the design, and there are likely to be many additional possibilities that we have not considered. With decades or perhaps centuries to work on the problem, the scientific community could work towards developing a plan that was both achievable and had a high probability of success.

Most of the research that needs to be done to move this process forward is research that we must do in order to predict future sea level rise anyway: coupled ice–ocean models; field studies of key glaciers; better understanding of basal hydrology, sediment transport, and erosion; oceanographic data from the sub–ice cavity; calving and fracture studies; and more. Glacial geoengineering is a dramatic topic that can capture popular interest (e.g. Meyer, 2018), providing a stimulus and popular appetite for more glaciological research. Glacial geoengineering also provides an additional set of questions that can inform the way we think about ice dynamics. How should the citizens of low–lying nations value ocean circulation in the sub–ice cavities of the Amundsen Sea? How much should the international community be willing to spend on the basal water pressure of important outlet glaciers? What exactly is the societal value of changes to the force balance of far away ice shelves? Geoengineering provides a framework for analyzing problems in glaciology that centers and quantifies the relationship between esoteric ice sheet processes and the concrete consequences of those processes for human societies and human lives.

The results that we have presented here are only the first step towards answering those questions. The designs we considered were very simple and our reduced dimensional model may miss important elements of the ice–ocean system. We only fully resolve one dimension in either the ice or the ocean, and we do not include any representation of ocean currents or mixing except in the ice–contact meltwater plume. However, in this case simplicity may be a virtue. Our ice model is mostly the same as the 1D model that Schoof used to define the modern theoretical understanding of the MISI (Schoof, 2007). The exact values





of collapse timing, sea level rise rate, and "point of no return" (the date at which an intervention would no longer be effective) will change with more advanced models, different forcings, and different intervention designs. The robust conclusions that can be drawn from our results are: 1) regrounding an ice shelf would slow an ongoing collapse, and 2) regrounding is more likely the more warm water is blocked from reaching the ice base. Neither of these two points is controversial (e.g. Joughin et al.,

2014; Seroussi et al., 2017), but taken together they suggest that consensus ice physics provide an opening for a large-scale civil engineering project to make a meaningful difference in the probability an ice sheet collapse.

One of the biggest potential failure points that must be addressed in future models is ice shelf disintegration caused by summer surface melt. The intervention we proposed relies on the buttressing force provided by the floating ice shelf in order to work, but surface meltwater damages the structural integrity of ice shelves and can cause them to disintegrate catastrophically,

as Larsen B did in 2002 (Scambos et al., 2003). However, some ice shelves are protected by surface rivers that efficiently export meltwater off the shelf (Bell et al., 2017). Future research is required to determine the extent to which surface meltwater reduces the probability of success for glacial geoengineering, to quantify how that probability reduction depends on atmospheric warming and hence on carbon emissions, and to determine whether it would be possible to deliberately modify supraglacial hydrology so as to encourage meltwater export.

Regardless of whether or not the intervention is successful, it is likely to have unintended consequences. One of the advantages of locally targeted geoengineering is that many of those unintended consequences are likely to also be local in nature. In the case of an artificial sill, changes to the local ocean circulation will be extensive by design, and turbidity will be increased during construction. Both of these are likely to have effects on marine biology. Not only must all side effects be addressed in detail before the sill could actually be built, but an additional set of moral and political questions must be addressed as well.

One of those questions is the issue of decision-making. The mass balance of Greenland and Antarctica affects nations around the globe, but no legal mechanism currently exists for deciding how humanity should go about trying to control those ice sheets. Antarctica is governed by the Antarctic Treaty, but the Greenland Ice Sheet is under the sovereign control of a specific nation, with a local population of 58,000 in a semi–autonomous relationship with Denmark (CIA, 2013). We don't know whether authority over geoengineering legally resides with Copenhagen or with Nuuk, but morally we do not believe

that geoengineering should proceed in Greenland without the consent of the Greenlandic people.

Another question is moral hazard, the risk that geoengineering may be used as a political argument to justify continued carbon emissions, and that research into it will therefore undermine climate mitigation. We could counter this by pointing out that the MISI may have already begun in the Amundsen sector (Joughin et al., 2014; Favier et al., 2014; Rignot et al., 2014). If that is so, then humanity will still have to deal with an ice sheet collapse even if we stopped all emissions tomorrow.

However, the point is moot if knowledge of geoengineering does not actually decrease people's support for climate mitigation, and empirical support for the moral hazard hypothesis within the social science literature is mixed (Burns et al., 2016). Properly contextualized discussion of geoengineering can actually *increase* concern for climate change (Kahan et al., 2015; Merk et al., 2016), consistent with other research demonstrating that positive, practical, or solution–based messaging is more effective at communicating climate science than negative, apocalyptic, or fear–based messaging (O'Neill and Nicholson-Cole, 2009;

Feinberg and Willer, 2011).





In addition, the denialist argument that carbon emissions are justified by geoengineering is wrong on the merits. Firstly, there are many harmful consequences of climate change in addition to rising sea levels, such as droughts, floods, heat waves, extreme weather, ocean acidification, and more (IPCC, 2014). Glacial geoengineering does nothing about these other threats, or even about sea level rise due to ocean thermal expansion. Secondly, atmospheric warming increases the production of summer

meltwater on floating ice shelves, which could lead the intervention to fail through shelf disintegration as discussed above. Thirdly, in a warming climate the collapse of other overdeepened basins in Antarctica becomes more likely (e.g. DeConto and Pollard, 2016), multiplying the number of interventions required. Finally, even if the interventions work as intended we still could not save the ice sheets indefinitely if humanity does not get emissions under control. On millennial timescales, the evolution of the ice sheets is controlled by cumulative $CO_2$ emissions (Winkelmann et al., 2015). In a strongly warming world,

the only viable long–term goal of glacial geoengineering is a managed collapse.

## 7    Conclusions

Many of us feel an understandable aversion to the thought of deliberately controlling the Earth's climate. Locally targeted interventions may offer a milder alternative to traditional large–scale geoengineering. Rather than trying to manage the entire planet, we could focus our intervention on specific high–leverage areas, like ice streams and outlet glaciers. This is not a

project that would begin soon. Field tests would be decades away at the earliest, and humanity might not be ready to deal with Thwaites for a century or so. In the short run our priority remains reducing emissions, because our emissions today will impact the climate for over a hundred thousand years (Keeling and Bacastow, 1977; Archer, 2005). But in the long run we need plans to deal with the committed climate changes that are already in the pipeline, one of which may be an ice sheet collapse (Joughin et al., 2014; Favier et al., 2014; Rignot et al., 2014). Those plans could include both traditional coastal protection and targeted

geoengineering. Managing sea level rise at the source has the advantage of benefiting the entire world equally, while a strategy that relies only on local coastal protection is more of an every-nation-for-itself approach that may leave many poor countries behind. The ideas that we have put forward here are only the beginning of a long incremental process of design improvement that will be necessary before the scientific community settles on the right plan. Perhaps, after careful consideration, we may conclude that glacial geoengineering is unworkable and the right answer is to spend heavily on coastal protection and retreat

inland where that is not practical. However, we owe it to the 400 million people who live within 5 m of sea level (Nicholls et al., 2008) to carefully consider the alternatives.

*Code availability.*   Model code available from the authors by request.

*Competing interests.*   The authors declare that they have no competing financial interests.



*Acknowledgements.* MW was supported under award NA14OAR4320106 from the National Oceanic and Atmospheric Administration, U.S. Department of Commerce. The statements, findings, conclusions, and recommendations are those of the authors and do not necessarily reflect the views of the National Oceanic and Atmospheric Administration, or the U.S. Department of Commerce. JM was supported by Chinese MOST grant 2015CB953602. Olga Sergienko provided helpful comments on earlier versions of the manuscript. Richard Alley provided

5  feedback on later versions.



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
