# Peer review of "Stopping the Flood: Could We Use Targeted Geoengineering to Mitigate Sea Level Rise?"

_The Cryosphere, 2018_

## Short Comment (SC1) · 13 Jun 2018

I first read the article Moore et al. 2018 in Nature and this sparked my interest in this proposal. This manuscript is focused on the proposal to build an artificial sill to protect the ice from melting caused by the warmer incoming water. The authors used a simplified model to investigate a time period of 1000 years based on different scenarios, which included sills with different blockage rates. Costs and feasibility are discussed and also lighter constrictions are mentioned. It is obvious a paper to use as a starting point and think/discuss the proposal further.

I have to notify that I'm not an expert in this research field but hope that my suggestions/questions further strengthen the understandability for a bigger audience.

- Abstract and page(P) 13 line(L) 15: The 30% probability of success is an estimate based on which calculation or boundary conditions?

- The water level is fixed for the calculation time because it is comparable small to the ice sheet. I couldn't find a comment on this in the discussion part.

- As above mentioned, I'm not an expert in this particular research field. Consequently, in my opinion it would be very useful if in the section Methods (or alternatively in the supporting information) the simplifications are summarised, which gives the reader a previous overview of the used model and its assumptions.

- P6 L10: The construction time of 10 years seems to be a very conservative approach but it's reasonable keeping the difficult boundary conditions as well as the length in mind. The fixed beginning of the work in 100 years should be questioned; especially, when looking at the animations. Except of the animation 5 all other scenarios would suggest that the sill is built under the floating ice shelf, which would lead to a dramatic increase of the difficulty and cost of the intervention. Maybe a flexible starting point of the work based on an ice-free sea on top of the potential cross section would be a good starting point.

- P9 L7: How did the authors define the collapse of the ice sheet? Which criteria was used? Please clarify this.

- Figure 5: It would be good to have a clear connection the individual subfigures and the added animations.

- The discussion starts with a general comment and some further (research) questions and ends with the geoengineering. In the middle part the used approach is discussed and ranked as a first step. I would suggest to split this into two different subsections. One with a general discussion and an additional part, in which all assumptions are summarised.
- The animation 1 and 2 only cover 120 years of the total investigated 1000 years. It would be nice to see the full period to compare it to those with the intervention. All animations show the similar first 100 years and in consequence the novelty of these videos are only 20 years.

- If I understand it correctly, the animation 1 and 2 show the same result hence the build structure has 0% blockage. It would be very useful, if in the section Movies in the supplement documents the choice of the presented movies would be explained and also the main findings summarised.

I would like to thank the authors for their very interesting paper and I'm looking forward to the final version. Thank you!

---

## Author Comment (AC1) · 14 Jun 2018

We thank Dr. Gabl for his comment on our article. We now respond to specific points below:

*Abstract and page(P) 13 line(L) 15: The 30% probability of success is an estimate based on which calculation or boundary conditions?*

The 30% number is based on the isolated pinning points scenario, in which the sill blocks none of the warm water but still provides buttressing should the ice shelf reground on it. Isolated pinning points is the easiest to build of the designs that we

considered, which is why we used the 30% number in these two locations. The abstract already clarified that the 30% number refers to the smallest design, and we have clarified the wording in the discussion as well.

***The water level is fixed for the calculation time because it is comparable small to the ice sheet. I couldn't find a comment on this in the discussion part.***

That is correct, we kept sea level constant throughout the simulation. Although there are many factors that contribute to the sea level budget, they are all relatively small compared with the bed depth below sea level (500-1500 m). The maximum potential sea level contribution from Greenland is 7 m, (Bamber et al., 2013), while smaller glaciers and ice caps provide less than a meter and thermal expansion may contribute 1-2 m over a millennial timescale (Levermann et al., 2013).

The potential sea level contribution from Thwaites itself and from the rest of Antarctica is more complex, since near-field sea level actually falls when an ice sheet loses mass due to gravitational effects. We expect that the sea level changes due to Thwaites itself and to the rest of Antarctica to be a stabilizing feedback potentially counteracting the destabilizing effect of mass loss in Greenland, consistent with the results of Gomez et al. (2010) that we cite in the introduction.

***As above mentioned, I'm not an expert in this particular research field. Consequently, in my opinion it would be very useful if in the section Methods (or alternatively in the supporting information) the simplifications are summarised, which gives the reader a previous overview of the used model and its assumptions.***

Thank you for pointing out that this wasn't necessarily clear to all readers. The beginning of the Methods (P4, L27) mentions that we use a flowband model, and the supplement mentions that we employ the Shallow Shelf Approximation (also called the Shelfy-Stream Approximation, both abbreviated SSA). What "flowband SSA" means is that we treat the flow of the glacier in a width- and depth-averaged sense. We have

added a bit of wording at the beginning of the Methods section clarifying this. We still present the model equations in detail in the supplement, but if other reviewers feel that a more lengthy list of model assumptions is needed in the main text, then we can add that too. However, for now we think that it is sufficient to mention the two biggest assumptions involved in a flowband SSA model, namely that the model is depth-averaged and width-averaged.

*P6 L10: The construction time of 10 years seems to be a very conservative approach but it's reasonable keeping the difficult boundary conditions as well as the length in mind. The fixed beginning of the work in 100 years should be questioned; especially, when looking at the animations. Except of the animation 5 all other scenarios would suggest that the sill is built under the floating ice shelf, which would lead to a dramatic increase of the difficulty and cost of the intervention. Maybe a flexible starting point of the work based on an ice-free sea on top of the potential cross section would be a good starting point.*

The way that we implemented sill construction in the model code required that the start date, location, sill dimensions, and construction duration all be specified before the model run began. In future work we could explore the effect of implementing a dynamic sill construction code that decides when to build the sill based on the condition of the model glacier, and potentially modifies the sill dimensions in response to the glacier geometry. This would also be a good place to test different levels of societal foresight: how does it change things if society begins construction when the glacier first starts retreating, as compared to a society that only starts building once the retreat has become very severe?

*P9 L7: How did the authors define the collapse of the ice sheet? Which criteria was used? Please clarify this.*

The model glaciers had a very bimodal response depending on whether they entered a runaway marine ice sheet collapse or not. For those that did not collapse, the grounding

line never strayed more than 10 km behind the present-day position and ice volume stayed close to the present-day value. For glaciers that did enter a runaway retreat, the grounding line retreated hundreds of kilometers inland, volume above flotation collapsed, and the rates of both retreat and volume loss increased drastically. In practice we defined a collapse as any run in which the grounding line retreated more than 25 km, however alternate thresholds based on volume loss, rate of retreat, or rate of volume loss would have produced similar results.

*Figure 5: It would be good to have a clear connection the individual subfigures and the added animations.*

All snapshots in Figure 5 come from animation 3. We have clarified this in the figure caption.

*The discussion starts with a general comment and some further (research) questions and ends with the geoengineering. In the middle part the used approach is discussed and ranked as a first step. I would suggest to split this into two different subsections. One with a general discussion and an additional part, in which all assumptions are summarised.*

We will consider this possible reorganization depending on what the reviewers say.

*The animation 1 and 2 only cover 120 years of the total investigated 1000 years. It would be nice to see the full period to compare it to those with the intervention. All animations show the similar first 100 years and in consequence the novelty of these videos are only 20 years.*

Those animations only have 120 years to show because the model became numerically unstable in the late stages of collapse, and we included code that automatically terminated the model run if numerical instability was detected. The final snapshot of those animations shows an extremely thin ice shelf extending far out into the sea; this ice shelf would have rapidly calved away in the ensuing year. Since our numerical scheme
always uses the same number of grid cells no matter the length of the glacier, a retreating calving front causes the grid cells to shrink, and since the timestep remains constant, this would eventually cause a numerical stability threshold to be crossed. Steep ice thickness gradients and fast ice velocities in the late stage of a collapse also lower the threshold for numerical instabilities to grow. Setting a timestep short enough to preserve numerical stability in the late stages of a collapse would have increased processing time in the rest of the model runs.

We did not prioritize the ability of our model to simulate the late stages of a collapse since the assumptions behind a flowband model break down in the late stages of a collapse anyway: width-averaged models assume that the cross-flow structure of the glacier remains the same over time, but large retreats should change the ice sheet geometry sufficiently to draw in substantial ice from the sides and destabilize neighboring glacier basins. We assume that Thwaites finishes collapsing after the model run ends, and brings the rest of West Antarctica with it (Feldmann and Levermann, 2015).

*If I understand it correctly, the animation 1 and 2 show the same result hence the build structure has 0% blockage. It would be very useful, if in the section Movies in the supplement documents the choice of the presented movies would be explained and also the main findings summarised.*

As mentioned in the methods section (P6, L13-14), we use the scenario with 0% water blocking to represent isolated artificial pinning points instead of a continuous artificial sill. In other words, the structure provides no water blockage but it does provide physical buttressing if the ice regrounds on it. Animations 1 and 2 show the same behavior because the ice shelf never regrounds, resulting in a failed intervention. However, if the shelf had been able to reground in animation 2 then the subsequent model evolution would have been different. We have added wording on P6 L14 to clarify this.

*I would like to thank the authors for their very interesting paper and I'm looking forward to the final version. Thank you!*

Thank you for your comment, and you're welcome!

**References**

Bamber, J. L., Griggs, J. A., Hurkmans, R. T. W. L., Dowdeswell, J. A., Gogineni, S. P., Howat, I., et al. (2013). A new bed elevation dataset for Greenland. The Cryosphere, 7(2), 499–510. https://doi.org/10.5194/tc-7-499-2013

Feldmann, J., Levermann, A. (2015). Collapse of the West Antarctic Ice Sheet after local destabilization of the Amundsen Basin. Proceedings of the National Academy of Sciences, 112(46), 14191. https://doi.org/10.1073/pnas.1512482112

Gomez, N., Mitrovica, J. X., Huybers, P., Clark, P. U. (2010). Sea level as a stabilizing factor for marine-ice-sheet grounding lines. Nature Geoscience, 3, 850.

Levermann, A., Clark, P. U., Marzeion, B., Milne, G. A., Pollard, D., Radic, V., Robinson, A. (2013). The multimillennial sea-level commitment of global warming. Proceedings of the National Academy of Sciences, 110(34), 13745–13750. https://doi.org/10.1073/pnas.1219414110

---

## Short Comment (SC2) · 15 Jun 2018

I want to thank the authors for their excellent answers and comments. Thank you!
* * *

---

## Referee Comment (RC1) · S.J. Marshall (Referee) · 24 Jul 2018

I will be brief in my comments, but mostly because I found this manuscript to be carefully considered and extremely well-presented. Congratulations to the authors for a very thoughtful contribution. I am disturbed that the world is coming to this - contemplation of what may seem to be fanciful, profligate civil engineering projects to try and influence something as monumental as an ice sheet. But I also agree with the authors that such interventions need to be discussed and considered. The world is arriving here, and sea-level rise on order of even 0.5 m (never mind 1 m or more) will be far more expensive and disruptive to society than what is presented here.

[Figure]

It is my overall opinion that the authors have a good understanding of the glaciological and technical considerations here, and are appropriately circumspect in their discussion. The language is neutral and cautious, and the implications and limitations of their study are thoroughly considered. I also believe that this is of broad interest to readers of TC and more generally in the field of climate change science, mitigation, adaptation, and policy, so this article is likely to be highly cited. I recommend it for publication in TC with only minor clarifications.

General suggestions, for consideration:

p.4, ll.7-9, discussion of rates of sea level rise. I don't think the rates that are cited are representative of the consensus of "modern models". Rates of several m per century are only really possible from Antarctica, in association with a marine-calving collapse, i.e. the ice-cliff instability of Pollard and de Conto. From the Thwaites system, ice resistive stresses and deformational velocities generally limit the rate if deglaciation, according to most model studies to date, and this will be true for most Antarctic embayments. In the example of the last deglaciation, the sea level rise of up to 5 m/century was in a much different world, with huge mid-latitude ice sheets capable of (surface) melt rates that are not possible in the polar regions. I think these examples are still fine to mention, but don't need to be considered as the "likely" scenario for the future centuries. Especially as rates of sea level rise of an order of magnitude less than this would still be massively disruptive and would justify potential interventions.

p.4,5, Methods. It is a little worrying that the model used for this study does not appear to consider longitudinal stresses. These are important to floating ice dynamics, grounding line migration, and the timescale of marine ice sheet instabilities. This should be discussed.

p.5, Experiments. Really interesting. I worry a bit that the interventions don't address the mechanical conditions that drive MISI - subglacial topography, stress balance, and pinning points upstream of the grounding line. I appreciate that warm water (basal

melting) strongly influences the ice thickness and then feeds back on these things, such that an ice readvance, if it can be triggered, can then bring the ice sheet back out to the manufactured sill, with possibilities to ground and stabilize. But I think there are some who would suggest that the MISI is a mechanical instability that is associated with the upstream geometry and, once triggered, it can continue without regard to ocean temperatures (i.e., with no need of enhanced melting). Again, a brief discussion of this could be helpful.

Are there oceanographic or 'storm' considerations here for effective blocking of threatening CDW by a sill? That is, are conditions so strongly stratified that a manufactured sill that does not completely block the embayment can effectively isolate the ice sheet? I don't know if tidal mixing or storm- or katabatic-driven Ekman fluxes, etc., can effectively mix the water column (especially in a future with less sea ice/a longer summer open water season), limiting the efficacy of the manufactured sills. But perhaps they just need to initially trigger ice thickening and advance, and then the mechanical grounding does the job.

And some minor comments:

abstract, l.9, "is both effective and achievable"

p.3, l.10, 1990s

p.3, l.18. displacement of 100-500 million people per year - I think this must be total, not per year. As this would not be a very sustainable rate of migration.

p.15, l.15. I am not sure that field tests could be decades away at the earliest - the authors argue that pilot tests in some Greenlandic fjords could be reasonable to contemplate. But point taken - we have time to develop more complete models and thoroughly consider oceanographic/marine biological considerations.
* * *

---

## Referee Comment (RC2) · X. Asay-Davis (Referee) · 24 Jul 2018

**Review of Wolovick and Moore (2018) "Stopping the Flood: Could We Use Targeted Geoengineering to Mitigate Sea Level Rise?"**

Reviewer: Xylar Asay-Davis

I wish my name to be relayed to the authors, as I do not support the practice of anonymous review.

**General Comments:**

The paper describes a set of 135 simulations with a simplified (1 horizontal dimension, 1HD) coupled ice sheet--ocean model designed to explore the implications of constructing a sill or a series of ice rises in the vicinity of Thwaites Glacier, West Antarctica. The simulations show that the intervention is increasingly successful depending on a combination of 1) how much warm water is blocked from entering the ice-shelf cavity and 2) the amount of additional buttressing provided to the ice shelf. The authors find an ~30% of reducing retreat with an isolated ice rise but much higher odds of reducing or reversing retreat with more extensive sills of various sizes and fractional widths.

The authors also discuss various practical, ethical and political questions that arise with regard to the feasibility and desirability of a geoengineering project along the lines they explore in this paper. These include the requirement that the technological capability would need to be proven and explored at smaller scales; that local environmental and societal impacts would need to be fully explored; and the possibility that other physical processes not captured in the simulations the authors performed, notably surface melting and subsequent ice-shelf breakup, might render the intervention ineffective.

I think the manuscript presents a compelling platform for further discussion of the possible benefits but also the challenges and potential unintended consequences of geoengineering related to ice sheets and glaciers. I think the manuscript is appropriate for publication in The Cryosphere but would like to see so revisions that are mostly minor, as detailed below.

My two most significant concerns about the work are the following. First, I am concerned that 1HD modeling is not appropriate for Thwaites Glacier because the complex topography and significant cross-flow variability are likely to provide buttressing that is fundamentally 2HD and cannot be captured through a 1HD parameterization (see detailed discussion below). I would have liked to see at least some validation of the 1HD approximation through comparison with 2HD modeling.

Second, the parameterization of ambient water masses in the ice-shelf cavity assumes that the properties of the deepest water masses in a partially obstructed cavity would be a linear combination (proportional to the fraction of obstruction) of those at the deepest point in the open ocean and those at the top of the sill that provides the partial obstruction. It is my assessment that ocean modeling and observations suggest that partial obstruction is not very efficient at blocking water masses form being transported horizontally. This would

suggest that the warmer, deeper water mass would likely fill the deeper parts of the cavity even when most (but not all) of the width of the cavity is blocked by a sill. For many ice shelves around Antarctica, troughs either near the continental shelf break or beneath the ice shelf itself provide efficient pathways for warm water to enter ice-shelf cavities even when these troughs represent only a small fraction of the width of the shelf. To me, this suggests that a re-interpretation of the results with 50% sill blockage may be required. Again, see details below.

**Specific Comments:**

p. 1 l. 2: "Thwaites Glacier, West Antarctica, is the largest individual source of future sea level rise". This needs to be reworded slightly, I think. You say later of Thwaites undergoing MISI, "We regard this hypothesis to be probable but not yet proven." It seems like the abstract could use similar qualification like "will likely be" or "is projected to be".

p. 1 l. 3 "coupled ice–ocean flowband simulations". In my experience, "flow band" is a meaningful term in 2D "side-view" ice-sheet modeling that parameterizes the 3rd dimension (e.g. Price et al. 2017, doi: 10.1029/2006JF000724) but it is not used in ocean modeling as far as I'm aware. So I would suggest coming up with a different term to describe the coupled model (2D; quasi-2D; 2D, side-view; or something like that).

Fig. 1: I rarely say this but I think some of the text may be too big in this figure. Particularly the titles of each panel seem too large. Also, you use uppercase letters for panels in Figs. 1, 2 and 5 but lowercase for Figs. 3 and 4. I much prefer lowercase (which seems to be standard) but more importantly would like to have consistent numbering

Fig. 2: I would leave a bit more space between each panel title and the panel itself. Also, I found it distracting that the titles seem to be in a different font from the other text (though this may just be an odd boldface font).

p. 3 l. 12-13: "There is also uncertainty about whether the ocean forcing that (may have) pushed the ice sheet over the edge was caused by human activity (Steig et al., 2012)" I would recommend citing a other papers that make this case more forcefully: Turner et al. 2017 DOI:10.1002/2016RG000532 (see Sec. 6. Attribution of Recent Changes in the ASE). The recent evident that Pine Island began its present retreat before the 1940s (Smith et al. 2016, DOI:10.1038/nature20136) might point to a lower likelihood that anthropogenic forcing played a role in that glacier's retreat.

p. 3 l. 13-15: "We proceed with the understanding that the societal consequences of a collapse will be the same regardless of whether or not humanity is responsible." This point is well stated.

p. 3 l. 17, p. 4 l. 2: I hate to keep pushing you to equivocate more but I would suggest changing "would" to something like "would, by some estimates". I know this is implied by the citations you give but with projections in general and cost estimates in specific it doesn't hurt to be explicit about what we know vs. what can only be an approximation.

p. 4 l. 17: Are other glaciers "less challenging" simply in being smaller, or are there other aspects that make Thwaites particularly challenging? If the latter, maybe mention something about these explicitly (or tell the reader you'll get to them later).

p. 4 l. 21: "merely piles of aggregate on the ocean floor". Would aggregate be strong enough to remain intact as the ice re-advances over it? Or might the artificial sill be weak and therefore short-lived? These are engineering challenges that are probably beyond the scope of this paper but they may figure into the feasibility if building an artificial sill strong enough to serve as an ice rise turns out to be cost-prohibitive.

p. 4 l. 27: "We use the least complex model that can address this question…" I get that you wanted to use a simple tool. I get, also, that it's kind of a first cut, a feasibility study. But I do wonder if the answers might not be totally different in a model that can fully represent buttressing and also the lateral variability of the topography. I guess I'm concerned that the model might be a little *too* simple to be able to give you a reliable answer to your questions. The flowband model is likely more prone to MISI (both is the sense of unstable retreat and unstable readvance) than a 3D model because of the fact that buttressing is parameterized as a drag or a change in viscosity. Furthermore, the nature of buttressing represented in a 1HD model is fundamentally different from that in a 2HD model (Gudmundsson et al. 2012 DOI: 10.5194/tc-6-1497-2012). Ideally, you would validate a few of your 135 model runs with a 2HD model. If that is too much to ask, I would suggest that you include here or in the discussion a thorough airing of these potential limitations of your 1HD model, in which much of the introduction and discussion material in Gudmundsson et al. (2012) is likely relevant.

p. 6. l. 15-16: "For the 50% blockage experiment, the ocean properties forcing the sill model were a linear combination of the properties at the sill top and the far–field stratification." Could you explain this choice further? Ocean dynamics is typically mostly horizontal, suggesting that the deepest water mass would flood the cavity for any percentage less than 100% sill blockage (assuming the percentage is meant to represent a horizontal fraction of the channel width that is covered by a sill). I do not think the the choice to have colder water in the cavity because a sill blocking 50% of the channel width is not consistent with observations or modeling of ocean dynamics in similar topographies. The warmer, denser water is perfectly content to flow around the obstacle and fill the region behind it, preventing the cooler, less dense water from descending over the sill to mix at depth. I think your 50% simulation is more representative of the behavior if you had a sill that was half as high (at least from the ocean's perspective) but covered the full width.

p. 7 l. 3-6: "The price of this feature is that our model cannot include the marine ice cliff instability, which could play an important role in accelerating West Antarctic collapse (DeConto and Pollard, 2016)." I didn't follow this argument. Are you saying that you wouldn't get accelerated calving for large cliffs because you would have a slow calving rate rather than a fast one for large H compared with $H_0$?

"However, this feature also guaranteed that our model never produced unphysically large ice cliffs in the first place, so in practice this was not an issue." Some in the field would dispute

the implication that MICI requires "unphysically large ice cliffs." While that may be true, I think wading into that particular controversy is beyond the scope of this paper and should probably be left out.

Over all, found these two sentences to be strange. You suggest you're missing a potentially important bit of calving physics if you encounter large ice cliffs but then dismiss it because your calving parameterization is such that you never do encounter large cliffs. Should we be relieved or does that just point to more potentially missing physics in your calving parameterization?

Fig 4: All fonts seem giant, but maybe this figure is meant to be smaller in the published version? As in Fig 2, the title font seems weird compared with the non-bold font and titles seem really close to the top of each panel.

p. 7 l. 30-33: These two sentences come as something of a non-sequitur. I presume the point is that you simply prescribed a change in the thermocline depth because you didn't feel you could derive changes from CMIP5 simulations. Even so, it's not clear where the justification for the 200-300 m shoaling comes from.

p. 7 l. 32: Another appropriate citation here would be Little and Urban (2016, DOI: 10.1017/aog.2016.25).

p. 9 l. 1-3: "For lower blocking percentages, the water properties behind the sill were a linear combination of the far–field stratification and the water properties at the sill top." Same complaint as on p. 6: This doesn't seem consistent with ocean dynamics.

Fig 6: I think both the y axis and the quantity being plotted in color need further explanation. Presumably the y axis is representing the percentage of model runs with that rate of sea level rise *or lower*, correct? Otherwise I really don't understand the y axis. Regarding the color map, is this the instantaneous rate the moment regrounding occurs? Or at the end of the 1000 year simulation? Or averaged over some time?

p. 12 l. 6-7: "With knowledge of the route of ocean currents in the sub-ice cavity, it may be possible to get the water–blocking performance of a continuous sill with less material." For the reasons I discussed above, this seems unlikely to me. Ocean water at depth is efficient at flowing around obstacles. It is energetically very favorable to flow along constant density surfaces and a partial blockage is unlikely to impede the flow or reduce the temperature of water in the cavity in a way that significantly reduces melting.

p. 13 l. 14-`5: "and it would have only a 30% probability of success" → "and our results suggest that it would…" or something along those lines.

p. 13 l. 24-25: "How should the citizens of low–lying nations value ocean circulation in the sub–ice cavities of the Amundsen Sea?" Perhaps the ambiguity is intentional but it is not

clear what you mean by "value". Do you mean monetary value (or at least a tangible value that can be monetized) or something more intangible and cultural, political or otherwise sociological?

p. 13 l. 24-25: "How much should the international community be willing to spend on the basal water pressure of important outlet glaciers?" I don't follow this question. Up until now, basal hydrology didn't figure into this discussion and it is not clear to me that there are any known or proposed interventions that would affect basal water pressure in a controlled way. So I am not aware of any way in which the international community could spend money on basal water pressure in any meaningful way. If the intention is to posit a fanciful means of further geoengineering ice sheets and glaciers, that probably needs to be made more explicit.

p. 13 l. 32-33: "However, in this case simplicity may be a virtue." I don't find that this case is made sufficiently to warrant this statement. Presumably the virtue is that you are able to perform well over 100 simulations with different model configurations. But I don't think the implications of these simplifications are sufficiently explored.

"Our ice model is mostly the same as the 1D model that Schoof used to define the modern theoretical understanding of the MISI (Schoof, 2007)." A lot of literature (notably Gudmundsson et al. 2012, mentioned above) has explored the limitations of the 1HD understanding of MISI as well as 1HD approximations of 2HD buttressing.

p. 14 l. 1-2: "The exact values of collapse timing, sea level rise rate, and "point of no return" (the date at which an intervention would no longer be effective) will change with more advanced models, different forcings, and different intervention designs." I think this sentence implies that differences between 1HD and 2HD modeling are likely to be in the small details. I don't think this is well established, and I would not be surprised to see qualitative changes in behavior (e.g. reduced MISI but also potentially increased difficulty re-advancing with new pinning points) with a 2HD model compared with the 1HD model used here. I feel like the tone of this sentence kind of undermines the point made just above that, "The designs we considered were very simple and our reduced dimensional model may miss important elements of the ice–ocean system."

p 14-15: I really appreciated this discussion of the political and ethical implications of this work. It is atypical of a paper in The Cryosphere but it a vital part of a discussion of a new potential geoengineering project.

p. 15: "Code availability. Model code available from the authors by request." Do you have a compelling reason for not making the code publicly available? If so, in my view, this should be state here. If not, I think the code should be made public (even if in an unsupported and perhaps poorly or undocumented form). I realize this is not the policy of The Cryosphere but I ask you to consider it anyway.

S3: I'm wondering how you handled "subglacial lakes" between two grounded regions that are visible in some of the animations in the supplementary material. Was there any melting in these regions? Hopefully not, since these regions presumably aren't actually supplied with heat from the ocean. Also, the plume would need to be re-initialized at each grounding line, which would be technically tricky.

**Typographical and grammatical corrections:**

p. 1 l. 2-3 and elsewhere: "the MISI" is typically just "MISI" in most texts I've read (just as it's not typically "the WAIS", though that would make grammatical sense). Obviously, this is a matter of taste.

p. 1 l. 3 "flowband" should probably be "flow band" or "flow-band" if you choose to retain this phrase.

p. 2 l. 5: "(MISI)(Fig 1)" would be cleaner as "(MISI; Fig 1)"

p. 3 l. 7: "West Antarctica(Joughin" missing a space before the parenthesis.

p. 4 l. 18-19: "The question that we seek to answer is…" Shouldn't this be, "The questions that we seek to answer are..."?

p. 4 l. 27: "this question" → "these questions"?

p. 6 l. 3: "supplementary section 1.3" should probably just be "S1.3" for consistency with the rest of the text.

Many places: phrases like "low–lying" and "sub–ice" are separated by en-dashes that should be normal dashes. (Presumably something the typesetter will handle.) This is as opposed to "ice–ocean", which arguably should have an en-dash.

---

## Author Comment (AC2) · 9 Aug 2018

**Author response to review by S.J. Marshall on "Stopping the Flood: Could We Use Targeted Geoengineering to Mitigate Sea Level Rise?"**

Michael Wolovick and John Moore

We thank Dr. Marshall for the positive review of our article. We now respond to specific comments below.

*p.4, ll.7-9, discussion of rates of sea level rise. I don't think the rates that are cited are representative of the consensus of "modern models". Rates of several m per century are only really possible from Antarctica, in association with a marine-calving collapse, i.e. the ice-cliff instability of Pollard and de Conto. From the Thwaites system, ice resistive stresses and deformational velocities generally limit the rate if deglaciation, according to most model studies to date, and this will be true for most Antarctic embayments. In the example of the last deglaciation, the sea level rise of up to 5 m/century was in a much different world, with huge mid-latitude ice sheets capable of (surface) melt rates that are not possible in the polar regions. I think these examples are still fine to mention, but don't need to be considered as the "likely" scenario for the future centuries. Especially as rates of sea level rise of an order of magnitude less than this would still be massively disruptive and would justify potential interventions.*

All three of the models we cite in this part (DeConto and Pollard, 2016; Winkelmann et al., 2015; and Golledge et al., 2015) predict rates of sea level rise from Antarctica of at least a meter per century under high emission scenarios. DeConto and Pollard (2016) do indeed have both the highest rate of sea level rise and the earliest peak (up to 6 m/century in the mid-2100's, shown in their Figure 4c), but the others are fairly large as well, and neither of the other two models included the marine ice cliff instability. Golledge et al. (2015) have the slowest rate of sea level rise of these three models; they show sea level rise rates in their Figure 2a that hit a maximum of about 15 mm/yr (1.5 m/century) around the year 2300. Winkelmann et al. (2015) do not show a figure depicting the rate of sea level rise; however, they do show cumulative sea level rise in their Figure 1d, and we were able to manually measure the derivative of those curves to determine that the highest emission scenario they consider had a sea level rise rate of 5.7 m/century between the years 2200 and 2300. We show our work for this calculation in the attached Figure 1 below.

In addition, the expert judgment assessment of Bamber and Aspinall (2013) shows that glaciologists believed (even before the 2014 papers hypothesizing the onset of the MISI in the Amundsen sector were published) that the 95[th] percentile for sea level rise in the year 2100 was 17.6 mm/yr, or 1.8 m/century. That expert elicitation produced a highly skewed probability distribution of sea level rise, and the authors explicitly connected the "fat tail" at the high end to experts allowing for the possibility that the MISI might be initiated in West Antarctica before the year 2100. Yet a runaway collapse, even if initiated before 2100, would probably not hit its maximum rate until the centuries after that. Considering those lags in the system, it is probably safe to say that the consensus opinion of the glaciological community is that sea level rise rates of greater than a meter per century are a reasonable expectation for a runaway ice sheet collapse.

It is true that the collapse of the mid-latitude ice sheets at the last deglaciation is a very different setting than a collapse of Antarctica would be in the future. However, we felt that it was important to cite data in this section in addition to models. The example of sea level rise rates during Meltwater Pulse 1a gives an indication of the order of magnitude of sea level rise rates that ice sheets are capable of during

a rapid collapse, and an argument based on both data and models is inherently stronger than an argument based on models alone. In addition, there is evidence that MWP1a may have been sourced from Antarctica rather than the mid-latitude ice sheets in the Northern Hemisphere (Clark et al., 2002). While the majority of the sea level rise that occurred during the last deglaciation was due to the melting of the Northern Hemisphere ice sheets, it is at least possible that the rapid rise during MWP1a was due to dynamic retreat in Antarctica, in which case this geologic evidence would be highly relevant. We have added wording in this part clarifying that the geologic evidence pertains to MWP1a specifically rather than the last deglaciation as a whole.

[Figure]

Figure 1. Our manual measurement of sea level rise rate from Winkelmann et al. (2015). The underlying figure is taken from Figure 1d of Winkelmann et al. (2015), showing cumulative sea level rise from Antarctica for a variety of emissions scenarios. We imported this image into LibreOffice Draw and manually overlaid regularly spaced horizontal and vertical lines in order to measure the slopes of the curves. The vertical lines are set at 100 yr increments and the horizontal lines are set at 1 m increments.

*p.4,5, Methods. It is a little worrying that the model used for this study does not appear to consider longitudinal stresses. These are important to floating ice dynamics, grounding line migration, and the timescale of marine ice sheet instabilities. This should be discussed.*

Our model does include longitudinal stresses; these are represented by the first term in Equation 2 of the supplementary material. We have changed the wording of the beginning of the methods section in the main text to clarify this.

*p.5, Experiments. Really interesting. I worry a bit that the interventions don't address the mechanical conditions that drive MISI - subglacial topography, stress balance, and pinning points upstream of the grounding line. I appreciate that warm water (basal melting) strongly influences the ice thickness and then feeds back on these things, such that an ice readvance, if it can be triggered, can then bring the ice sheet back out to the manufactured sill, with possibilities to ground and stabilize.*

This is a good point. One of the other interventions that we suggested in Moore et al. (2018) was a subglacial drying scheme designed to modify the stress balance upstream of the grounding line. Society will have to consider a wide variety of possible interventions before anything could actually be implemented, but for this particular paper we wanted to focus on evaluating the efficacy of one particular intervention. We leave it to future work to evaluate the relative merits of an artificial sill as compared to other potential interventions.

*But I think there are some who would suggest that the MISI is a mechanical instability that is associated with the upstream geometry and, once triggered, it can continue without regard to ocean temperatures (i.e., with no need of enhanced melting). Again, a brief discussion of this could be helpful.*

The MISI is indeed a mechanical instability, and it can be suppressed by the buttressing provided by a floating ice shelf (Gudmundsson et al., 2012). That is why we only considered an intervention to have been a success if the ice shelf regrounded on the artificial sill. Merely reducing the melt rate of an unbuttressed ice shelf makes no difference to the MISI, although for other glaciers whose shelves are in confined embayments, such as Pine Island Glacier, thickening the shelf will increase buttressing and slow grounding line retreat. However, for an unconfined shelf like Thwaites, reducing the basal melt rate of the shelf will only have an effect on the MISI if the shelf thickens enough that it regrounds on the sill.

*Are there oceanographic or 'storm' considerations here for effective blocking of threatening CDW by a sill? That is, are conditions so strongly stratified that a manufactured sill that does not completely block the embayment can effectively isolate the ice sheet? I don't know if tidal mixing or storm- or katabatic-driven Ekman fluxes, etc., can effectively mix the water column (especially in a future with less sea ice/a longer summer open water season), limiting the efficacy of the manufactured sills. But perhaps they just need to initially trigger ice thickening and advance, and then the mechanical grounding does the job.*

We have not explicitly considered ocean currents or mixing other than in the ice-contact meltwater plume. We use the scenario where the sill blocked 50% of the warm water to represent partial mixing of warm water over the sill top. As we discuss in our response to Dr. Asay-Davis' review, that scenario was not meant to represent 50% horizontal blockage, but rather full horizontal blockage with some of the water nonetheless being mixed over the sill top by tides/winds/storms etc. We have clarified our intentions with respect to this scenario in the methods section.

*abstract, l.9, "is both effective and achievable"*

Changed.

*p.3, l.10, 1990s*

Changed.

*p.3, l.18. displacement of 100-500 million people per year - I think this must be total, not per year. As this would not be a very sustainable rate of migration*

That number refers to temporary displacements due to episodic flooding and storms. We have modified this sentence to include both temporary and permanent population displacements. We had not initially included a number for permanent displacements since most of the literature we consulted considered the no-protection scenario to be unrealistically apocalyptic and they therefore did not quote a number for coastal refugee flows in the absence of coastal protection. We approximated the number for $21^{st}$ century sea level rise by taking the number of people within 1m of sea level (131 million, Nicholls et al., 2008) and dividing by 100 years. Nicholls et al. (2008) also give a more rigorous result for permanent refugee flows in the presence of coastal protection, and found permanent population displacements of tens to hundreds of thousands of people per year depending on the scenario. We have included both of these numbers in that paragraph.

*p.15, l.15. I am not sure that field tests could be decades away at the earliest - the authors argue that pilot tests in some Greenlandic fjords could be reasonable to contemplate. But point taken - we have time to develop more complete models and thoroughly consider oceanographic/marine biological considerations*

We have changed these sentences from, "This is not a project that would begin soon.
Field tests would be decades away at the earliest, and humanity might not be ready to deal with Thwaites for a century or so." to, "This is not a project that would begin soon. A large amount of modelling, data collection, planning, technological/logistical development, and field testing, not to mention public discussion and political debate, must be done first. Humanity might not be ready to deal with Thwaites for a century or so. "

**References**
Bamber, J. L., & Aspinall, W. P. (2013). An expert judgement assessment of future sea level rise from the ice sheets. Nature Clim. Change, 3(4), 424–427. https://doi.org/10.1038/nclimate1778
Clark, P. U., Mitrovica, J. X., Milne, G. A., & Tamisiea, M. E. (2002). Sea-level fingerprinting as a direct test for the source of global meltwater pulse IA. *Science*, *295*(5564), 2438–41.
DeConto, R. M., & Pollard, D. (2016). Contribution of Antarctica to past and future sea-level rise. *Nature*, *531*(7596), 591–597. https://doi.org/10.1038/nature17145
Golledge, N. R., Kowalewski, D. E., Naish, T. R., Levy, R. H., Fogwill, C. J., & Gasson, E. G. W. (2015). The multi-millennial Antarctic commitment to future sea-level rise. *Nature*, *526*(7573), 421–425. https://doi.org/10.1038/nature15706
Gudmundsson, G. H., Krug, J., Durand, G., Favier, L., & Gagliardini, O. (2012). The stability of grounding lines on retrograde slopes. *The Cryosphere*, *6*(6), 1497–1505. https://doi.org/10.5194/tc-6-1497-2012
Moore, J. C., Gladstone, R., Zwinger, T., & Wolovick, M. J. (2018). Geoengineer polar glaciers to slow sea-level rise. *Nature*, *555*, 303–305. https://doi.org/10.1038/d41586-018-03036-4
Nicholls, R. J., Tol, R. S. J., & Vafeidis, A. T. (2008). Global estimates of the impact of a collapse of the West Antarctic ice sheet: an application of FUND. *Climatic Change*, *91*(1), 171.

https://doi.org/10.1007/s10584-008-9424-y

Winkelmann, R., Levermann, A., Ridgwell, A., & Caldeira, K. (2015). Combustion of available fossil fuel resources sufficient to eliminate the Antarctic Ice Sheet. *Science Advances, 1*(8). https://doi.org/10.1126/sciadv.1500589

---

## Author Comment (AC3) · 9 Aug 2018

**Author response to review by X. Asay-Davis on "Stopping the Flood: Could We Use Targeted Geoengineering to Mitigate Sea Level Rise?"**

Michael Wolovick and John Moore

We thank Dr. Asay-Davis for the thorough review of our article. We now respond to specific comments below.

*My two most significant concerns about the work are the following. First, I am concerned that 1HD modeling is not appropriate for Thwaites Glacier because the complex topography and significant cross-flow variability are likely to provide buttressing that is fundamentally 2HD and cannot be captured through a 1HD parameterization (see detailed discussion below). I would have liked to see at least some validation of the 1HD approximation through comparison with 2HD modeling.*

We feel that 1HD modeling is actually more appropriate for Thwaites than for other glaciers, since Thwaites is so wide that side drag plays a very small role in its dynamics. In addition, while there is some cross-flow variability in Thwaites' basal topography, there is no well-defined central trough or confined ice shelf as in many other glaciers; and it was specifically the presence of a central trough and a confined ice shelf that Gudmundsson et al. (2012) used to generate the lateral buttressing that can stabilize a glacier against the MISI. We expand more on this point below.

*Second, the parameterization of ambient water masses in the ice-shelf cavity assumes that the properties of the deepest water masses in a partially obstructed cavity would be a linear combination (proportional to the fraction of obstruction) of those at the deepest point in the open ocean and those at the top of the sill that provides the partial obstruction. It is my assessment that ocean modeling and observations suggest that partial obstruction is not very efficient at blocking water masses form being transported horizontally. This would suggest that the warmer, deeper water mass would likely fill the deeper parts of the cavity even when most (but not all) of the width of the cavity is blocked by a sill. For many ice shelves around Antarctica, troughs either near the continental shelf break or beneath the ice shelf itself provide efficient pathways for warm water to enter ice-shelf cavities even when these troughs represent only a small fraction of the width of the shelf. To me, this suggests that a re-interpretation of the results with 50% sill blockage may be required. Again, see details below*

We had not intended the 50% blockage experiment to represent 50% horizontal blockage, but rather 50% mixing of the water over the top of the sill. It is absolutely true that ocean currents are efficiently transported horizontally. We intended that experiment to represent a situation in which a sill was constructed across the entire width of the bay, but because of winds/tides/internal waves/storms/etc some of the warm water was mixed over the top of the sill. Perhaps it would have been more realistic to parameterize these processes by using a uniform water mass in the cavity behind the sill composed of a mixture between the warm deep waters and the cold shallow waters; however, we chose to preserve some of the far-field stratification since that represented a more stringent test of the effectiveness of the sill. By preserving some of the far-field stratification, we ensured that the deep water reaching the grounding line was warmer than it would have been if we had filled the cavity with a uniform water mass.

In the aggregate volume calculations in Table 1 we did not consider any sills that partially covered the width of the bay; the only partial horizontal coverage we considered was the case of isolated pinning points, in which case we assumed 0% water blockage. The different aggregate volumes we calculated

for continuous sills were entirely due to different assumptions about sill position (whether in the wide open bay or on the narrower high bathymetry near the present-day grounding line), sill height (with the sill top either 300 m, 250 m, or 100 m below the surface) and aggregate strength (with an angle of repose of 15° or 45°). Both the 50% blockage experiment and the first 100% blockage experiment use the same assumed sill geometry, corresponding to superscript (3) in Table 1. That design is described as a low sill built on the higher bathymetry near the present-day grounding line. The assumption behind that design geometry is that the grounding line would have retreated to form a large embayment before construction begins. The mouth of the embayment would then form a natural constriction (relatively speaking; the length of the sill is still 80 km) roughly at the location of the present-day grounding line, and since the present-day grounding line also has the highest bathymetry in the area, building a sill there would be doubly favored. We did not actually run any experiments corresponding to (4) (low sill in the open bay), we only included that sill geometry calculation in the table as an example of a smaller open-bay design. The most effective scenario we considered (tall sill in the open bay) was (5) in Table 1.

We have added wording in the experiment description section (3.1) to clarify our interpretation of the 50% blockage experiment. We have also added wording to the caption of Table 1 clarifying which scenarios correspond to which designs.

*p. 1 l. 2: "Thwaites Glacier, West Antarctica, is the largest individual source of future sea level rise". This needs to be reworded slightly, I think. You say later of Thwaites undergoing MISI, "We regard this hypothesis to be probable but not yet proven." It seems like the abstract could use similar qualification like "will likely be" or "is projected to be".*

We have added the qualifier "is projected to be".

*p. 1 l. 3 "coupled ice–ocean flowband simulations". In my experience, "flow band" is a meaningful term in 2D "side-view" ice-sheet modeling that parameterizes the 3rd dimension (e.g. Price et al. 2017, doi: 10.1029/2006JF000724) but it is not used in ocean modeling as far as I'm aware. So I would suggest coming up with a different term to describe the coupled model (2D; quasi-2D; 2D, side-view; or something like that).*

We have changed the word, "flowband" to "quasi-2D".

*Fig. 1: I rarely say this but I think some of the text may be too big in this figure. Particularly the titles of each panel seem too large. Also, you use uppercase letters for panels in Figs. 1, 2 and 5 but lowercase for Figs. 3 and 4. I much prefer lowercase (which seems to be standard) but more importantly would like to have consistent numbering*

We did not know that it was possible to have the text in a figure too big!

We also do not know how big this figure will be after final typesetting, and this cartoon is a good candidate for getting squeezed into a single column when the article gets typeset into two columns, so we prefer to leave the text large. However, we have switched to lowercase lettering for the panels in Figs 1, 2, and 5.

*Fig. 2: I would leave a bit more space between each panel title and the panel itself. Also, I found it distracting that the titles seem to be in a different font from the other text (though this may just be an odd boldface font).*

The title font is the same, just bold.  We have moved the titles slightly up and away from the panels.

*p. 3 l. 12-13: "There is also uncertainty about whether the ocean forcing that (may have) pushed the ice sheet over the edge was caused by human activity (Steig et al., 2012)" I would recommend citing a other papers that make this case more forcefully: Turner et al. 2017 DOI:10.1002/2016RG000532 (see Sec. 6. Attribution of Recent Changes in the ASE). The recent evident that Pine Island began its present retreat before the 1940s (Smith et al. 2016, DOI:10.1038/nature20136) might point to a lower likelihood that anthropogenic forcing played a role in that glacier's retreat.*

We have added both of these references.

*p. 3 l. 13-15: "We proceed with the understanding that the societal consequences of a collapse will be the same regardless of whether or not humanity is responsible." This point is well stated.*

Thank you.

*p. 3 l. 17, p. 4 l. 2: I hate to keep pushing you to equivocate more but I would suggest changing "would" to something like "would, by some estimates". I know this is implied by the citations you give but with projections in general and cost estimates in specific it doesn't hurt to be explicit about what we know vs. what can only be an approximation.*

We have changed the word, "would" to, "could" in order to imply more uncertainty.

*p. 4 l. 17: Are other glaciers "less challenging" simply in being smaller, or are there other aspects that make Thwaites particularly challenging? If the latter, maybe mention something about these explicitly (or tell the reader you'll get to them later).*

We were mostly thinking of size here, but the severely overdeepened geometry of Thwaites without a stabilizing topographic trough or substantial ice-shelf buttressing also contribute to the difficulty.  In addition, the fact that the MISI may have already been triggered in the Amundsen Sea Embayment adds an additional degree of difficulty, in that humanity may be working against the clock when it comes to developing the technological and logistical capabilities necessary to stage an intervention.  We have added a sentence discussing these factors.

*p. 4 l. 21: "merely piles of aggregate on the ocean floor". Would aggregate be strong enough to remain intact as the ice re-advances over it? Or might the artificial sill be weak and therefore short-lived? These are engineering challenges that are probably beyond the scope of this paper but they may figure into the feasibility if building an artificial sill strong enough to serve as an ice rise turns out to be cost-prohibitive.*

These are issues to be explored in future work.  We have done some experiments with sill erosion that suggest that a weak sill could still be effective in delaying an ice sheet collapse.  However, those rely on an arbitrary erosion parameterization, and without some sort of calibration we do not consider those results to be meaningful.

*p. 4 l. 27: "We use the least complex model that can address this question…" I get that you wanted to use a simple tool. I get, also, that it's kind of a first cut, a feasibility study. But I do wonder if the answers might not be totally different in a model that can fully represent buttressing and also the*

*lateral variability of the topography. I guess I'm concerned that the model might be a little too simple to be able to give you a reliable answer to your questions. The flowband model is likely more prone to MISI (both is the sense of unstable retreat and unstable readvance) than a 3D model because of the fact that buttressing is parameterized as a drag or a change in viscosity. Furthermore, the nature of buttressing represented in a 1HD model is fundamentally different from that in a 2HD model (Gudmundsson et al. 2012 DOI: 10.5194/tc-6-1497-2012). Ideally, you would validate a few of your 135 model runs with a 2HD model. If that is too much to ask, I would suggest that you include here or in the discussion a thorough airing of these potential limitations of your 1HD model, in which much of the introduction and discussion material in Gudmundsson et al. (2012) is likely relevant.*

It is true that a flowband model is incapable of truly representing the full 2HD dynamics of buttressing. However, the geometry that Gudmundsson et al. (2012) considered was a very specific one: an ice stream confined to a deep central trough, connected to a laterally confined ice shelf, with higher ground on either side. The central trough served to confine both the fast-flowing trunk of the ice stream and the floating ice shelf that formed once the centerline ungrounded. It is important to emphasize that *all* of the lateral buttressing in the Gudmundsson model came from the gradient in ice velocity between the elevated flanks and the depressed central trough. The side walls of their model domain were free slip boundaries that did not provide any drag to the flow (see section 2, "Problem Definition", in that paper). As a result, the only part of their model ice shelf that was laterally buttressed was the area that was confined by grounded ice on the elevated flanks. Further downstream, where the ice on the flanks also ungrounded, the ice shelf was completely unbuttressed and contributed no significant resistance to flow.

The "central trough and confined shelf" geometry explored by Gudmundsson et al. (2012) is a very common geometry for ice streams and outlet glaciers, so it was a sensible choice for that study. The findings of their study have obvious implication for many overdeepened ice streams and outlet glaciers, including several in the Amundsen Sea Embayment for which the onset of the MISI has been hypothesized. Pine Island, Pope, Smith, and Kohler Glaciers all have a bedrock trough and a confined ice shelf, so the findings of Gudmunsson et al. (2012) would caution against extrapolating from the recent retreats of those glaciers to the conclusion that an irreversible collapse has begun.

However, Thwaites has neither a central trough nor a confined shelf. Though it does have some cross-flow variability, it does not have the specific structures shown by Gudmundsson to stabilize an ice stream against the MISI. Our simple force balance inversions (Fig 3c) suggest that side drag is negligible in the force balance of Thwaites compared to driving stress and basal drag. Those results are broadly consistent with more complex inversions which show that the rather high driving stress of Thwaites is balanced by local (or near-local) basal drag (Joughin et al., 2009; Morlighem et al., 2013). Even the inversion of Sergienko and Hindmarsh (2013), which probably has the most non-local stress transmission of any of the Thwaites inversions, does not show stress transmission to the margins, but rather to a specific pattern of sticky patches within the ice stream itself. The Sergienko and Hindmarsh (2013) model has a balance between driving stress and basal drag when considered at wavelengths longer than the spacing between the sticky patches, and that spacing (~11 km) is much less than the width of the glacier. One of the underlying assumptions of the canonical 1HD description of the MISI by Schoof (2007) is that basal drag and driving stress are balanced in the inland region of the ice stream, with only a small boundary region near the grounding line where longitudinal stresses are important as well. Our model represents all of those terms, and if there is any glacier in the world to which the Schoof (2007) description truly applies, that glacier is Thwaites.

Nonetheless, in a real retreat of Thwaites Glacier it is likely that different areas of the ice stream would

retreat at different rates, and this asynchronous retreat would create embayments in which a well-buttressed ice shelf could form. Such temporary buttressing would probably slow the retreat, since it would preferentially apply buttressing to the areas that are retreating the fastest, and that sort of process is not something that we can represent in a 1HD model.

We have added wording to the methods section elaborating on the weaknesses of a 1HD model and commenting on the relationship between Thwaites' geometry and the geometry considered by Gudmundsson et al. (2012).

*p. 6. l. 15-16: "For the 50% blockage experiment, the ocean properties forcing the sill model were a linear combination of the properties at the sill top and the far–field stratification." Could you explain this choice further? Ocean dynamics is typically mostly horizontal, suggesting that the deepest water mass would flood the cavity for any percentage less than 100% sill blockage (assuming the percentage is meant to represent a horizontal fraction of the channel width that is covered by a sill). I do not think the the choice to have colder water in the cavity because a sill blocking 50% of the channel width is not consistent with observations or modeling of ocean dynamics in similar topographies. The warmer, denser water is perfectly content to flow around the obstacle and fill the region behind it, preventing the cooler, less dense water from descending over the sill to mix at depth. I think your 50% simulation is more representative of the behavior if you had a sill that was half as high (at least from the ocean's perspective) but covered the full width.*

It is absolutely correct that the ocean currents should flow around a horizontal obstacle. As mentioned above, we intended the 50% blockage experiment to represent a case where the sill was built completely across the width of the bay, but was only partially effective at preventing transport of the warm water due to winds, tides, storms, internal waves, and other sources of variability in the thermocline depth. We have added wording to clarify our interpretation of this experiment.

*p. 7 l. 3-6: "The price of this feature is that our model cannot include the marine ice cliff instability, which could play an important role in accelerating West Antarctic collapse (DeConto and Pollard, 2016)." I didn't follow this argument. Are you saying that you wouldn't get accelerated calving for large cliffs because you would have a slow calving rate rather than a fast one for large H compared with $H_0$?*

Yes, we were trying to say that our model would produce a low calving rate rather than a large one when H is much larger than $H_0$. We have clarified the wording here.

*"However, this feature also guaranteed that our model never produced unphysically large ice cliffs in the first place, so in practice this was not an issue." Some in the field would dispute the implication that MICI requires "unphysically large ice cliffs." While that may be true, I think wading into that particular controversy is beyond the scope of this paper and should probably be left out.*
*Over all, found these two sentences to be strange. You suggest you're missing a potentially important bit of calving physics if you encounter large ice cliffs but then dismiss it because your calving parameterization is such that you never do encounter large cliffs. Should we be relieved or does that just point to more potentially missing physics in your calving parameterization?*

This is a good point. We have removed the second sentence.

*Fig 4: All fonts seem giant, but maybe this figure is meant to be smaller in the published version? As in Fig 2, the title font seems weird compared with the non-bold font and titles seem really close to the top*

*of each panel.*

We have moved the titles slightly away from the panels. We want to leave the font size large, however, as we do not know how big the figure will be after final typesetting.

*p. 7 l. 30-33: These two sentences come as something of a non-sequitur. I presume the point is that you simply prescribed a change in the thermocline depth because you didn't feel you could derive changes from CMIP5 simulations. Even so, it's not clear where the justification for the 200-300 m shoaling comes from.*

The 200-300 m shoaling is an arbitrary choice. The recent increases in sub-shelf melt and associated grounding line retreat in the Amundsen Sea Embayment have been caused by upwelling of warm CDW onto the continental shelf and associated increases of warm water transport into the sub-ice cavities, rather than substantial warming of the water mass itself. We wanted to create a forcing for the warming scenario that mimicked and magnified this trend, thus increasing the odds of collapse and making it harder to reverse the collapse with an intervention. We felt that the 200-300 m shoaling was a good order of magnitude for a plausible change in the destabilizing direction, but we wanted to put caveats at this part of the text emphasizing the uncertainty in actual projections of ocean circulation changes on the Amundsen shelf and especially in the sub-ice cavities. We have added wording here explicitly stating that the choice of 200-300 m was arbitrary.

*p. 7 l. 32: Another appropriate citation here would be Little and Urban (2016, DOI: 10.1017/aog.2016.25).*

We have added this reference.

*p. 9 l. 1-3: "For lower blocking percentages, the water properties behind the sill were a linear combination of the far–field stratification and the water properties at the sill top." Same complaint as on p. 6: This doesn't seem consistent with ocean dynamics.*

See our earlier responses.

*Fig 6: I think both the y axis and the quantity being plotted in color need further explanation. Presumably the y axis is representing the percentage of model runs with that rate of sea level rise or lower, correct? Otherwise I really don't understand the y axis. Regarding the color map, is this the instantaneous rate the moment regrounding occurs? Or at the end of the 1000 year simulation? Or averaged over some time?*

Before plotting, we sorted the model runs in order of post-regrounding sea level rise rate, so that is in fact the correct interpretation of the y-axis. The post-regrounding sea level rise rate shown in color is determined by the slope of a linear least squares fit of the time series between the time of regrounding and the end of the 1000 year model run. We have added wording to the caption clarifying these points.

*p. 12 l. 6-7: "With knowledge of the route of ocean currents in the sub-ice cavity, it may be possible to get the water–blocking performance of a continuous sill with less material." For the reasons I discussed above, this seems unlikely to me. Ocean water at depth is efficient at flowing around obstacles. It is energetically very favorable to flow along constant density surfaces and a partial blockage is unlikely to impede the flow or reduce the temperature of water in the cavity in a way that significantly reduces melting.*

We have removed this sentence. Instead, we have added a reference to the subglacial drying proposed by Moore et al. (2018). This reference works better in this position as an example of an alternative glacial geoengineering technique that could be explored, and also provides context for our mention of basal water pressure later in the discussion.

*p. 13 l. 14-`5: "and it would have only a 30% probability of success" → "and our results suggest that it would…" or something along those lines.*

We have qualified this sentence.

*p. 13 l. 24-25: "How should the citizens of low–lying nations value ocean circulation in the sub–ice cavities of the Amundsen Sea?" Perhaps the ambiguity is intentional but it is not clear what you mean by "value". Do you mean monetary value (or at least a tangible value that can be monetized) or something more intangible and cultural, political or otherwise sociological?*

We were originally thinking only of monetary/material value, but now that you point it out we quite like the ambiguity of interpretation that is possible here. We have reworded the next sentence from, "How much should the international community be willing to spend on the basal water pressure of important outlet glaciers?" to, "How much importance should the international community place on the basal water pressure of key outlet glaciers?" in order to ensure that the entire section can now be read with many meanings for "value".

*p. 13 l. 24-25: "How much should the international community be willing to spend on the basal water pressure of important outlet glaciers?" I don't follow this question. Up until now, basal hydrology didn't figure into this discussion and it is not clear to me that there are any known or proposed interventions that would affect basal water pressure in a controlled way. So I am not aware of any way in which the international community could spend money on basal water pressure in any meaningful way. If the intention is to posit a fanciful means of further geoengineering ice sheets and glaciers, that probably needs to be made more explicit.*

In a previous Nature Comment in which we proposed research into glacial geoengineering (Moore et al., 2018), subglacial drying was one of the ideas that we proposed as a potential avenue of research. We did not investigate that method in this paper, but this particular sentence was meant to refer to other potential intervention techniques. We do not think that our paper is the right place to get into detail about many alternative intervention techniques, but since this paragraph is discussing the merits of glacial geoengineering research on a very broad level we felt that it was appropriate to include examples of things other than the specific sills and pinning points that we considered in this paper. We have added a reference to subglacial drying in the "Cost and Feasibility" section in response to an earlier comment above, so hopefully this sentence does not appear to come out of nowhere anymore.

*p. 13 l. 32-33: "However, in this case simplicity may be a virtue." I don't find that this case is made sufficiently to warrant this statement. Presumably the virtue is that you are able to perform well over 100 simulations with different model configurations. But I don't think the implications of these simplifications are sufficiently explored.*

We have removed this sentence.

*"Our ice model is mostly the same as the 1D model that Schoof used to define the modern theoretical*

*understanding of the MISI (Schoof, 2007)." A lot of literature (notably Gudmundsson et al. 2012, mentioned above) has explored the limitations of the 1HD understanding of MISI as well as 1HD approximations of 2HD buttressing.*

We have also removed this sentence. We have replaced it, and the previous sentence, with: "Our model is the simplest model that can capture the mechanics of the MISI; indeed, it is mostly the same as the 1D model that Schoof (2007) used to define the modern theoretical understanding of the MISI. More advanced ice and ocean models are needed to fully explore lateral buttressing and ocean circulation in the sub-ice cavity."

As we mentioned above, Thwaites has neither a central trough nor a confined ice shelf, which are the two geometrical features which Gudmundsson used to get around the MISI in 2HD. We did include a parameterization of lateral buttressing in our model, but that term was only a small component of the force balance (Figure 3c). The buttressing that our intervention relies on is not lateral buttressing but rather longitudinal buttressing, which is fully represented in our model. We have mentioned the Gudmundsson caveat, along with the geometrical differences between Thwaites and the idealized glacier that they considered, in the methods section.

*p. 14 l. 1-2: "The exact values of collapse timing, sea level rise rate, and "point of no return" (the date at which an intervention would no longer be effective) will change with more advanced models, different forcings, and different intervention designs." I think this sentence implies that differences between 1HD and 2HD modeling are likely to be in the small details. I don't think this is well established, and I would not be surprised to see qualitative changes in behavior (e.g. reduced MISI but also potentially increased difficulty re-advancing with new pinning points) with a 2HD model compared with the 1HD model used here. I feel like the tone of this sentence kind of undermines the point made just above that, "The designs we considered were very simple and our reduced dimensional model may miss important elements of the ice–ocean system."*

We have added "success probability" to the list of things that might change with future work, but otherwise we are leaving this sentence as is. A reduced likelihood of unstable retreat due to the MISI, or a slower rate of retreat once the collapse is initiated, are both covered by changes in collapse timing, sea level rise rate, and the "point of no return". Similarly, slower recovery and increased difficulty re-advancing are covered under (post-regrounding) sea level rise rate and success probability.

*p 14-15: I really appreciated this discussion of the political and ethical implications of this work. It is atypical of a paper in The Cryosphere but it a vital part of a discussion of a new potential geoengineering project.*

Thank you.

*p. 15: "Code availability. Model code available from the authors by request." Do you have a compelling reason for not making the code publicly available? If so, in my view, this should be state here. If not, I think the code should be made public (even if in an unsupported and perhaps poorly or undocumented form). I realize this is not the policy of The Cryosphere but I ask you to consider it anyway.*

You are quite right, there was no good reason for us not to post the code online. It is now at github.com/MichaelWolovick/Flowline_v1.

*S3: I'm wondering how you handled "subglacial lakes" between two grounded regions that are visible in some of the animations in the supplementary material. Was there any melting in these regions? Hopefully not, since these regions presumably aren't actually supplied with heat from the ocean. Also, the plume would need to be re-initialized at each grounding line, which would be technically tricky.*

There was no melting in the "subglacial lakes". The plume started from the outermost grounding line. The freshwater forcing that initialized the plume was derived only from the outermost grounded region.

**Typographical and grammatical corrections:**

*p. 1 l. 2-3 and elsewhere: "the MISI" is typically just "MISI" in most texts I've read (just as it's not typically "the WAIS", though that would make grammatical sense). Obviously, this is a matter of taste.*

Grammatical conventions around acronyms are weird. People also say things like, "ATM machine" and "PIN number" and they sound correct even though those expressions are redundant. In most places where we say "the MISI" it would be inappropriate to just use "MISI" alone: or example, "The hypothesis that the MISI has already been triggered in the Amundsen sector..." makes no sense as, "The hypothesis that MISI has already been triggered...", but would sound okay if we said, "The hypothesis that a MISI has already been triggered...". For now we would prefer to leave it as "the MISI" since that actually makes sense if you expand the acronym into "the Marine Ice Sheet Instability".

*p. 1 l. 3 "flowband" should probably be "flow band" or "flow-band" if you choose to retain this phrase.*

We have replaced "flowband" with "flow-band". We removed the term from the abstract in response to a previous comment, but we have left the term in the main text.

*p. 2 l. 5: "(MISI)(Fig 1)" would be cleaner as "(MISI; Fig 1)"*

Changed.

*p. 3 l. 7: "West Antarctica(Joughin" missing a space before the parenthesis.*

Fixed.

*p. 4 l. 18-19: "The question that we seek to answer is…" Shouldn't this be, "The questions that we seek to answer are..."?*

Fixed.

*p. 4 l. 27: "this question" → "these questions"?*

Changed.

*p. 6 l. 3: "supplementary section 1.3" should probably just be "S1.3" for consistency with the rest of the text.*

Fixed.

*Many places: phrases like "low–lying" and "sub–ice" are separated by en-dashes that should be normal dashes. (Presumably something the typesetter will handle.) This is as opposed to "ice–ocean", which arguably should have an en-dash.*

Fixed.

**References**

Gudmundsson, G. H., Krug, J., Durand, G., Favier, L., & Gagliardini, O. (2012). The stability of grounding lines on retrograde slopes. *The Cryosphere*, *6*(6), 1497–1505. https://doi.org/10.5194/tc-6-1497-2012

Joughin, I., Tulaczyk, S., Bamber, J. L., Blankenship, D., Holt, J. W., Scambos, T., & Vaughan, D. G. (2009). Basal conditions for Pine Island and Thwaites Glaciers, West Antarctica, determined using satellite and airborne data. *Journal of Glaciology*, *55*(190), 245–257. https://doi.org/10.3189/002214309788608705

Moore, J. C., Gladstone, R., Zwinger, T., & Wolovick, M. J. (2018). Geoengineer polar glaciers to slow sea-level rise. *Nature*, *555*, 303–305. https://doi.org/10.1038/d41586-018-03036-4

Morlighem, M., Seroussi, H., Larour, E., & Rignot, E. (2013). Inversion of basal friction in Antarctica using exact and incomplete adjoints of a higher-order model. *Journal of Geophysical Research: Earth Surface*, *118*(3), 1746–1753. https://doi.org/10.1002/jgrf.20125

Schoof, C. (2007). Ice sheet grounding line dynamics: Steady states, stability, and hysteresis. *Journal of Geophysical Research- Earth Surface*, *112*(F3). https://doi.org/10.1029/2006JF000664

Sergienko, O. V., & Hindmarsh, R. C. A. (2013). Regular patterns in frictional resistance of ice-stream beds seen by surface data inversion. *Science*, *342*(6162), 1086–1089. https://doi.org/10.1126/science.1243903

---

## Author Response (AR2)

Author Response to Editor's Comments on "Stopping the Flood: Could We Use Targeted Geoengineering to Mitigate Sea Level Rise?" by Michael J. Wolovick and John C. Moore

*"the MISI" or "MISI": I agree with the reviewer that it would be more appropriate to use "MISI" without the definite article. While I concede that language norms are not always universal or logical, dropping the definite article would be most consistent with European style guides (see, e.g., http://publications.europa.eu/code/en/en-4100800en.htm), as well as modern usage of the term (e.g., Pattyn, 2018, Nature). Please adjust the text accordingly.*

It is your prerogative as the editor to insist on this change, and I have therefore removed the definite article from "MISI". However, I would like to register my objection for the record. **It just sounds wrong.**

*"Flowband" or "flow band": I believe either formulation of this term is acceptable. However, if you use the latter (as in the current version of the manuscript), then please only use the hyphenated form when it serves as an adjective "flow-band model" and otherwise "flow band" when it serves as a noun. Please check the manuscript for consistency here.*

If "flowband" (one word) is acceptable, then I would prefer to use that form. I only changed it at the request of the reviewer. I have therefore changed it back from two words to one word.

*Footnote on page 4: There is a typo here "The idea that society could face such severe disruption without doing •itsomething about it was deemed politically impossible.". Also please rephrase "deemed politically impossible" to the active sense to be consistent with the text – otherwise this sounds like it was deemed so by someone else.*

Both fixed.

*Figure 5, caption: also shown => is also shown*

Fixed.

*Page 14, line 25: "several decades" sounds speculative here. Please justify or rephrase (even just with the addition of a "will likely require").*

Fixed.

*Page 17, line 5: This phrase is also not easily justifiable: "This is not a project that would begin soon." I suggest deleting this sentence and adjusting the last part of the subsequent sentence: "must be done first" => "
[revised manuscript text omitted]